# The Advancement in Stochastic Zeroth-Order Optimization: Mechanism of Accelerated Convergence of Gaussian Direction on Objectives with Skewed Hessian Eigenvalues

## Abstract

This paper primarily investigates large-scale finite-sum optimization problems, which are particularly prevalent in the big data era. In the field of zeroth-order optimization, stochastic optimization methods have become essential tools. Natural zeroth-order stochastic optimization methods are primarily based on stochastic gradient descent (`SGD`). The method of preprocessing the stochastic gradient with Gaussian vector is referred to as `ZO-SGD-Gauss` (`ZSG`), while estimating partial derivatives along coordinate directions to compute the stochastic gradient is known as `ZO-SGD-Coordinate` (`ZSC`). Compared to `ZSC`, `ZSG` often demonstrates superior performance in practice. However, the underlying mechanisms behind this phenomenon remain unclear in the academic community. To the best of our knowledge, our work is the first to theoretically analyze the potential advantages of `ZSG` compared to `ZSC`. Unlike the fundamental assumptions applied in general stochastic optimization analyses, the quadratic regularity assumption is proposed to generalize the smoothness and strong convexity to the Hessian matrix. This assumption allows us to incorporate Hessian information into the complexity analysis. When the objective function is quadratic, the quadratic regularity assumption reduces to the second-order Taylor expansion of the function, and we focus on analyzing and proving the significant improvement of `ZSG`. For other objective function classes, we also demonstrate the convergence of `ZSG` and its potentially better query complexity than that of `ZSC`. Finally, experimental results on both synthetic and real-world datasets substantiate the effectiveness of our theoretical analysis.

## 1 Introduction

Modern machine learning presents significant challenges for optimization due to the large scale of the problems involved. Contemporary datasets are both enormous and high-dimensional, often with millions of samples and features. Because evaluating the full objective or gradient even once is too slow to be useful, stochastic optimization methods have emerged in response.

Throughout the paper, we aim to solve finite-sum minimization problems of the form

$$\min_{\boldsymbol{x} \in \mathbb{R}^d} f(\boldsymbol{x}) \stackrel{def}{=} \frac{1}{n} \sum_{i=1}^{n} f_i(\boldsymbol{x}). \tag{1}$$

An optimization method that solves the problem (1) with function value access only is known as zeroth-order optimization or black-box optimization (Ghadimi & Lan, 2013; Nesterov & Spokoiny, 2017). In recent years, zeroth-order optimization has attracted widespread attention from both the machine learning community and the optimization community (Nesterov & Spokoiny, 2017; Ilyas et al., 2018). One important application of the zeroth-order optimization is the black-box adversarial attack on deep neural networks (Chen et al., 2017; Zhao et al., 2020; Zhang et al., 2020; Bai et al., 2023). In the black-box adversarial attack, only the inputs and outputs of the neural network are available and back propagation is often prohibited (Papernot et al., 2017). In the above situation,

the evaluation of gradient $\nabla f(\boldsymbol{x})$ is infeasible. So, applying zeroth-order optimization methods becomes a natural choice. Additional application scenarios in the field of artificial intelligence where zeroth-order optimization algorithms demonstrate significant effectiveness are deep reinforcement learning (Salimans et al., 2017; Mania et al., 2018; Zhang & Zavlanos, 2023; Jing et al., 2024), hyper-parameter tuning (Snoek et al., 2012; Rapin & Teytaud, 2018), the problem of optimizing functions with only ranking feedback (Tang et al., 2023), learning linear quadratic regulators (Malik et al., 2020; Mohammadi et al., 2020), and so on. Zeroth-order optimization has even played a significant role in fine-tuning large language models (LLMs). Malladi et al. (2023) and Zhao et al. (2024) use the zeroth-order optimization methods for fine-tuning, in addressing the significant memory overhead of first-order optimizers. Zeroth-order optimization achieves a substantial memory reduction and makes it possible to train and store LLMs on low-cost hardware.

Though `ZO-SGD-Gauss` (ZSG) and `ZO-SGD-Coordinate` (ZSC) share the same theoretical convergence rate and their sample complexity is both linear to the dimension (Ghadimi & Lan, 2013), ZSG has wider application ranges and performs better than ZSC in practice. For example, ZSG has been widely used in fine tuning LLMs (Malladi et al., 2023; Zhao et al., 2024) and black-box attacks (Ilyas et al., 2018). The academic community is still unclear about the underlying mechanism why ZSG outperforms ZSC. For the gradient descent method, recent works by Yue et al. (2023) and Wang et al. (2024) show that zeroth-order Gaussian gradient descent can outperform coordinate descent if the Hessian has skewed eigenvalue distribution. An intriguing question is whether the zeroth-order SGD algorithm possesses a similar property to the zeroth-order gradient descent algorithm. Inspired by these works, we try to prove that ZSG can outperform ZSC under similar conditions. We obtain a surprising result: compared to ZSC, ZSG possesses weak dimensional dependence. Our work fills a theoretical gap in the field of zeroth-order optimization.

## 1.1 LITERATURE REVIEW

Here, we present a concise overview of stochastic optimization methods.

An optimization method that solves the problem (1) by accessing gradient information from a subset of samples is called SGD. SGD and its variance reduction variants, which operate on only a small mini-batch of data at each iteration, have become the preferred methods (Robbins & Monro, 1951; Moulines & Bach, 2011; Johnson & Zhang, 2013; Allen-Zhu, 2018). However, stochastic optimizers sacrifice stability in favor of speed. Parameters such as the learning rate are challenging to choose (Nemirovski et al., 2009), and for ill-conditioned large-scale machine learning problems, even finding the optimal learning rate can lead to very slow convergence. Second-order optimizers based on the Hessian, such as Newton's method (Battiti, 1992) and quasi-Newton methods (Dennis & Moré, 1977; Jin & Mokhtari, 2023), are the classic remedy for solving above challenges. Some researchers have proposed using stochastic Hessian approximations while still utilizing the full gradient (Lacotte et al., 2021; Tong et al., 2021). Then, Frangella et al. (2022) propose the `SketchySGD` algorithm whose excellent performance suggests it could potentially replace SGD.

When the gradient is difficult to calculate or cannot be obtained, researchers shift their attention from the study of SGD to stochastic zeroth-order optimization algorithms, estimating the gradient using function value differences (Ghadimi & Lan, 2013; Duchi et al., 2015; Nesterov & Spokoiny, 2017). Malladi et al. (2023) directly use zeroth-order optimizer (ZOO) for fine-tuning LLMs. However, the zeroth-order optimization algorithms mentioned above overlook the use of higher-order information about the objective, leading to less competitive convergence in practice. Similar to the development of SGD, researchers have begun to introduce second-order Hessian information into zeroth-order optimization algorithms. This idea holds promise for the design of efficient and competitive algorithms. Chen et al. (2017) utilize the second-order Hessian information in a relatively coarsened manner. Ye et al. (2018) take a first step to efficiently incorporate second-order Hessian information of the objective function and propose a novel class of algorithms called the ZOHA algorithm. Zhao et al. (2024) propose `HiZOO`, which is the first work to leverage the diagonal Hessian to enhance ZOO for fine-tuning LLMs.

It is worth noting that Nesterov & Spokoiny (2017) conduct a theoretical analysis of the complexity bounds for three random gradient-free oracles. However, they don't find the conditions under which ZSG algorithm performs better than ZSC algorithm. The essential reason is that they do

not effectively utilize the information from the Hessian matrix in their theoretical analysis process. Therefore, in essence, our work is different from that of (Nesterov & Spokoiny, 2017).

## 1.2 CONTRIBUTIONS

The main contributions of this paper are summarized as follows:

- Our work theoretically analyzes the conditions under which the ZSG algorithm outperforms the ZSC algorithm. To the best of our knowledge, our conclusion is innovative and novel, particularly in the context of quadratic functions. Through rigorous theoretical analysis, we demonstrate that as long as $\mathrm{tr}(\mathbf{M}) \ll d\lambda_{\max}(\mathbf{M})$, where $\mathbf{M}$ is the Hessian matrix of the objective function, the performance of the ZSG algorithm will surpass that of the ZSC algorithm.

- Our theoretical analysis is based on the upper and lower quadratic regularity assumptions, which generalize the $L$-smooth assumption and $\mu$-strongly convexity assumption. Building on a comprehensive theoretical examination of quadratic functions, we extend our complexity analysis conclusions to a broader class of functions.

- Our research indicates that ZSG also possesses weak dimensional dependence, similar to zeroth-order gradient descent. This fills a theoretical gap in the field of zeroth-order optimization, and our analytical results provide significant theoretical insights.

- Extensive experiments confirm the reliability of theoretical analysis of our work. Either using synthetically designed data or real-world datasets, the performance of the ZSG algorithm outperforms that of the ZSC algorithm.

## 2 NOTATION AND ASSUMPTIONS

Let us define the weighted Euclidean norm and weighted inner product associated with a positive definite weight matrix $\mathbf{M} \succ 0$

$$\|\boldsymbol{x}\|_{\mathbf{M}} \stackrel{def}{=} \langle \boldsymbol{x}, \boldsymbol{x} \rangle_{\mathbf{M}}^{\frac{1}{2}},$$

$$\langle \boldsymbol{x}, \boldsymbol{y} \rangle_{\mathbf{M}} \stackrel{def}{=} \langle \mathbf{M}\boldsymbol{x}, \boldsymbol{y} \rangle.$$

We define the stochastic gradient $\nabla f(\boldsymbol{x}, \mathcal{S}) = \frac{1}{|\mathcal{S}|} \sum_{j \in \mathcal{S}} \nabla f_j(\boldsymbol{x})$, where $\mathcal{S}$ represents the sample set and $|\mathcal{S}|$ represents the sample size.

A widely accepted notion is that the assumptions of $f$ being $L$-smooth and $\mu$-strongly convex are standard in the analysis of stochastic gradient methods for solving the problem (1). As the study of stochastic algorithms deepens, many researchers have proposed more generalized assumptions. Hanzely et al. (2018) introduce the $\mathbf{M}$-smoothness assumption, which is a common assumption in modern analyses of stochastic coordinate descent methods, to analyze the convergence of the SEGA algorithm and propose the Q-smoothness assumption, which further generalizes the $\mathbf{M}$-smoothness assumption. Gower et al. (2019) introduce the relative smoothness assumption and relative convexity assumption to exploit the information from the Hessian matrix. Frangella et al. (2023) utilize the quadratic regularity assumption to overcome the dilemma of infrequent preconditioner updates. Frangella et al. (2022) propose the relative quadratic regularity assumption, which replaces the Hessian matrix with any positive definite matrix.

Then, we make the following assumptions for the objective function $f$. First, we introduce the quadratic regularity assumption (Frangella et al., 2023), which can be viewed as a global generalizations of the smoothness and strong convexity constants to the Hessian norm.

**Assumption 2.1.** Let $f : \mathbb{R}^d \to \mathbb{R}$ be a twice differentiable function, and let $\mathbf{M}$ denote the Hessian matrix of $f$. The function $f$ is said to be upper quadratically regular with respect to $\mathbf{M}$ if, for all $\boldsymbol{x}, \boldsymbol{y}$ and $\boldsymbol{z} \in (\boldsymbol{x}, \boldsymbol{y})$, there exists a constant $0 < \gamma_u < \infty$ such that the following inequality holds:

$$f(\boldsymbol{y}) \le f(\boldsymbol{x}) + \langle \nabla f(\boldsymbol{x}), \boldsymbol{y} - \boldsymbol{x} \rangle + \frac{\gamma_u}{2} \|\boldsymbol{y} - \boldsymbol{x}\|_{\mathbf{M}(\boldsymbol{z})}^2. \tag{2}$$

Similarly, $f$ is said to be lower quadratically regular with respect to $\mathbf{M}$ if, for all $\boldsymbol{x}, \boldsymbol{y}$ and $\boldsymbol{z} \in (\boldsymbol{x}, \boldsymbol{y})$, there exists a constant $0 < \gamma_l < \infty$ such that the following inequality holds:

$$f(\boldsymbol{y}) \geq f(\boldsymbol{x}) + \langle \nabla f(\boldsymbol{x}), \boldsymbol{y} - \boldsymbol{x} \rangle + \frac{\gamma_l}{2} \|\boldsymbol{y} - \boldsymbol{x}\|^2_{\mathbf{M}(\boldsymbol{z})}. \tag{3}$$

We define the quadratic regularity ratio to be

$$q \overset{def}{=} \frac{\gamma_u}{\gamma_l}.$$

Frangella et al. (2023) also prove that $\gamma_u, \gamma_l$ and $q$ are independent of the condition number of the data for many popular machine learning problems.

Next, we introduce the standard variance control assumption.

**Assumption 2.2.** The variance of the stochastic gradient can be bounded by $\sigma^2$, which means

$$\mathbb{E}\left[\|\nabla f(\boldsymbol{x}, \mathcal{S}) - \nabla f(\boldsymbol{x})\|^2\right] \leq \sigma^2. \tag{4}$$

Rearranging above formula, we can obtain,

$$\mathbb{E}\left[\|\nabla f(\boldsymbol{x}, \mathcal{S})\|^2\right] \leq \|\nabla f(\boldsymbol{x})\|^2 + \sigma^2. \tag{5}$$

## 3 ALGORITHM DESCRIPTION

This section commences with a detailed description of the algorithm. In the following, we briefly introduce the classical zeroth-order gradient estimator SPSA (Spall, 1992).

**Definition 3.1.** (Simultaneous Perturbation Stochastic Approximation). Given a model with parameters $\boldsymbol{x} \in \mathbb{R}^d$ and loss function $f$, SPSA estimates the gradient on a minibatch $\mathcal{S}$ as

$$\hat{\nabla} f(\boldsymbol{x}, \mathcal{S}) = \frac{[f(\boldsymbol{x} + \alpha \boldsymbol{u}, \mathcal{S}) - f(\boldsymbol{x} - \alpha \boldsymbol{u}, \mathcal{S})]}{2\alpha} \cdot \boldsymbol{u} \approx \boldsymbol{u} \boldsymbol{u}^\top \nabla f(\boldsymbol{x}, \mathcal{S}), \tag{6}$$

where $\boldsymbol{u} \in \mathbb{R}^d$ is sampled from $\mathcal{N}(\mathbf{0}, \mathbf{I}_d)$ and $\alpha$ is a very small perturbation scale.

It should be noted that $\hat{\nabla} f(\boldsymbol{x}, \mathcal{S})$ is called the zeroth-order-estimated first-order gradient information. In order to help us prove complexity, we need to find the connection between the zeroth-order oracles and the gradient.

**Lemma 3.2.** *We access to the $f(\boldsymbol{x} + \alpha \boldsymbol{u}, \mathcal{S})$ and $f(\boldsymbol{x} - \alpha \boldsymbol{u}, \mathcal{S})$. Through the upper quadratically regular assumption, we yield the following equivalence relation*

$$\hat{\nabla} f(\boldsymbol{x}, \mathcal{S}) = \boldsymbol{u} \boldsymbol{u}^\top \nabla f(\boldsymbol{x}, \mathcal{S}) + \phi(\boldsymbol{u}, \alpha, \boldsymbol{x}), \tag{7}$$

*with*

$$\|\phi(\boldsymbol{u}, \alpha, \boldsymbol{x})\| \leq \frac{\gamma_u \alpha}{2} \|\boldsymbol{u}\|^2_{\mathbf{M}(\boldsymbol{z})} \cdot \|\boldsymbol{u}\|, \tag{8}$$

*where $\boldsymbol{z}_1 \in (\boldsymbol{x}, \boldsymbol{x} + \alpha \boldsymbol{u})$, $\boldsymbol{z}_2 \in (\boldsymbol{x} - \alpha \boldsymbol{u}, \boldsymbol{x})$ and $\mathbf{M}(\boldsymbol{z}) = \begin{cases} \mathbf{M}(\boldsymbol{z}_1) & \text{if } \mathbf{M}(\boldsymbol{z}_1) \succeq \mathbf{M}(\boldsymbol{z}_1) \\ \mathbf{M}(\boldsymbol{z}_2) & \text{otherwise} \end{cases}$.*

The detailed proof is presented in B.1. The aforementioned relationships can help us conduct convergence analysis. This paper focuses on analyzing the convergence properties of the following update rule:

$$\boldsymbol{x}^{t+1} = \boldsymbol{x}^t - \eta_t \hat{\nabla} f(\boldsymbol{x}^t, \mathcal{S}_t). \tag{9}$$

The main algorithmic procedure of the ZSG is provided in Algorithm 1.

---

**Algorithm 1** ZSG: ZO-SGD-Gauss Method

---

**Input and Initialize:** parameters $\boldsymbol{x} \in \mathbb{R}^d$, loss function $f : \mathbb{R}^d \to \mathbb{R}$, step budget $t$, step size $\eta_t > 0$, perturbation scale $\alpha$, sample distribution $\mathcal{D}$, initial point $\boldsymbol{x}^0 \in \mathbb{R}^d$
   **for** $t = 0, 1, \cdots$ **do**
      Sample $\mathcal{S}_t \sim \mathcal{D}$ and $\boldsymbol{u}_t \sim \mathcal{N}(\mathbf{0}, \mathbf{I}_d)$
      Query the zeroth-order oracle $f_+^t = f(\boldsymbol{x}^t + \alpha \boldsymbol{u}_t, \mathcal{S}_t)$
      Query the zeroth-order oracle $f_-^t = f(\boldsymbol{x}^t - \alpha \boldsymbol{u}_t, \mathcal{S}_t)$
      Estimating the gradient $\hat{\nabla} f(\boldsymbol{x}^t, \mathcal{S}_t) = \frac{(f_+^t - f_-^t)}{2\alpha} \cdot \boldsymbol{u}_t$
      $\boldsymbol{x}^{t+1} = \boldsymbol{x}^t - \eta_t \hat{\nabla} f(\boldsymbol{x}^t, \mathcal{S}_t)$
   **end for**

---

## 4 MAIN THEORETICAL RESULTS

This section provides an in-depth examination of the iterative complexity of ZSG under the assumptions we introduced. First, we study the convergence properties of quadratic functions. To explain the superiority of ZSG conveniently, we assume that $f(\boldsymbol{x}) = \frac{1}{2} \boldsymbol{x}^\top \mathbf{M} \boldsymbol{x} - \boldsymbol{b}^\top \boldsymbol{x}$. If the objective function $f$ in Assumption 2.1 is quadratic function, we need to point that $\gamma_l = \gamma_u = 1$ and $\mathbf{M}(\boldsymbol{z}) \equiv \mathbf{M}$, meaning the Hessian matrix is independent of the iteration points.

We begin by presenting several essential lemmas that help us derive the main theorems in this section. The detailed proofs of Lemma 4.1 and Lemma 4.2 are provided in Section B. In addition, several other lemmas and the proofs of them are listed in Section A. The detailed proofs of the main theorems and corollaries in this section are presented in Section C.

**Lemma 4.1.** *Let $\boldsymbol{u}_t \sim \mathcal{N}(\mathbf{0}, \mathbf{I}_d)$ be a random vector and $\boldsymbol{x} \in \mathbb{R}^d$ be an arbitrary point. For all $t>0$, the variance of the zeroth-order-estimated first-order gradient information can be bounded as follows:*

$$\mathbb{E}_{\boldsymbol{u}_t} \left[ \left\| \boldsymbol{u}_t \boldsymbol{u}_t^\top \nabla f(\boldsymbol{x}^t, \mathcal{S}_t) \right\|_{\mathbf{M}}^2 \right] \leq 3 \text{tr}(\mathbf{M}) \left\| \nabla f(\boldsymbol{x}^t, \mathcal{S}_t) \right\|^2. \tag{10}$$

**Lemma 4.2.** *Let $f^*$ be optimum of the objective function. For all $t>0$, if $\boldsymbol{z} \in (\boldsymbol{x}^t, \boldsymbol{x}^*)$, the difference between the function value at $\boldsymbol{x}^t$ and the optimum $f^*$ can be bounded as follows:*

$$f(\boldsymbol{x}^t) - f^* \leq \frac{1}{2\gamma_l} \left\| \nabla f(\boldsymbol{x}^t) \right\|_{\mathbf{M}(\boldsymbol{z})^{-1}}^2 \leq \frac{1}{2\gamma_l \lambda_{\min}(\mathbf{M}(\boldsymbol{z}))} \left\| \nabla f(\boldsymbol{x}^t) \right\|^2. \tag{11}$$

**Theorem 4.3.** *Let $f$ be quadratic function, and assume that $f$ is upper quadratically regular and lower quadratically regular with respect to $\mathbf{M}$. That is, Assumption 2.1 holds. In addition, the stochastic gradient is limited by the noise. That is, Assumption 2.2 holds. Let $\boldsymbol{x}^{t+1}$ be updated according to Eq. (9). We define $P_1(\alpha^2) = \frac{[\lambda_{\max}(\mathbf{M}) + 2\eta](6+d)^3 \alpha^2}{4 \lambda_{\min}(\mathbf{M})}$. We choose*

$$\eta_t \equiv \eta \leq \frac{1}{12 \text{tr}(\mathbf{M})}, \tag{12}$$

*then, we can obtain*

$$\mathbb{E} \left[ f(\boldsymbol{x}^{t+1}) - f^* \right] \leq \frac{6\eta \text{tr}(\mathbf{M}) \sigma^2}{\lambda_{\min}(\mathbf{M})} + P_1(\alpha^2) + \left[ 1 - \frac{1}{2} \eta \lambda_{\min}(\mathbf{M}) \right]^t \left[ f(x^0) - f^* \right].$$

We can observe that ZSG converges to a ball around the optimum from Theorem 4.3 when we choose fixed step size. This phenomenon is analogous to the classic SGD which employs a fixed learning rate (Moulines & Bach, 2011).

**Corollary 4.4.** *We observe Theorem 4.3 and find that if a fixed step size is chosen, the algorithm will eventually fail to converge in the presence of noise. Let $f$ satisfy the properties described in Theorem 4.3 and select the parameters described in Theorem 4.3. Since we can choose a sufficiently small $\alpha$ in practice, we can omit it. If $\sigma^2 = 0$, to find an $\varepsilon$-suboptimal solution, the iteration complexity is*

$$t = \mathcal{O} \left( \frac{\text{tr}(\mathbf{M})}{\lambda_{\min}(\mathbf{M})} \log \frac{1}{\varepsilon} \right). \tag{13}$$

When $\sigma^2 = 0$, that is, the update of $x$ depends on the real gradient, and we obtain the same conclusion as in (Wang et al., 2024). The conclusion of (Wang et al., 2024) is an intermediate analysis product of our work, and their purpose is to compare it with the coordinate sketching version of the SEGA (Hanzely et al., 2018) whose iteration complexity is $\mathcal{O}\left(\frac{d\lambda_{\max}(\mathbf{M})}{\lambda_{\min}(\mathbf{M})}\log\frac{1}{\varepsilon}\right)$ and achieve the best convergence rate without the importance sampling. Obviously, when condition $\text{tr}(\mathbf{M}) \ll d\lambda_{\max}(\mathbf{M})$ is met, this algorithm is better than SEGA algorithm. However, our work focus on the analysis of stochastic gradient. The following theorem and corollary will indicate that ZSG outperforms ZSC.

**Theorem 4.5.** *Let f be quadratic function, and suppose that Assumption 2.1 and Assumption 2.2 hold. Let $\boldsymbol{x}^{t+1}$ be updated according to Eq. (9). Then, we let general $\eta_t = \frac{l}{\gamma+t}$ be decreasing and $\gamma > 0$. In addition, we assume that $t_{\max} = T$ and we can fix the intermediate parameter $l = \frac{3}{\lambda_{\min}(\mathbf{M})}$. We define $Q_1(\alpha^2) = \frac{[18+108\lambda_{\max}(\mathbf{M})\text{tr}(\mathbf{M})+3\lambda_{\max}(\mathbf{M})\lambda_{\min}(\mathbf{M})T](6+d)^3\alpha^2}{4\lambda_{\min}^2(\mathbf{M})}$. We choose initial step size*

$$\eta_0 = \frac{l}{\gamma} \leq \frac{1}{12\text{tr}(\mathbf{M})}, \tag{14}$$

*which means we can obtain a lower bound for parameter $\gamma$,*

$$\gamma \geq \frac{36\text{tr}(\mathbf{M})}{\lambda_{\min}(\mathbf{M})}. \tag{15}$$

*Then, we choose another parameter*

$$v = \max\left\{\gamma(f(\boldsymbol{x}^0) - f^*), \frac{54\text{tr}(\mathbf{M})\sigma^2}{\lambda_{\min}^2(\mathbf{M})} + Q_1(\alpha^2)\right\}. \tag{16}$$

*Finally, we can obtain*

$$\mathbb{E}\left[f(\boldsymbol{x}^t) - f^*\right] \leq \frac{v}{\gamma+t}. $$

**Corollary 4.6.** *We observe Theorem 4.5 and find that if a decreasing step size is chosen in practice, the algorithm will eventually converge in the presence of noise. Let f satisfy the properties and select the parameters described in Theorem 4.5. The following holds: to find an $\varepsilon$-suboptimal solution, the iteration complexity is*

$$t = \mathcal{O}\left(\left[\frac{\text{tr}(\mathbf{M})\sigma^2}{\lambda_{\min}^2(\mathbf{M})} + Q_1(\alpha^2)\right]\frac{1}{\varepsilon}\right). \tag{17}$$

When $\sigma^2 > 0$ and a sufficiently small $\alpha$ is chosen in practice, the iteration complexity of Algorithm 1 is $\mathcal{O}\left(\frac{\text{tr}(\mathbf{M})\sigma^2}{\lambda_{\min}^2(\mathbf{M})}\frac{1}{\varepsilon}\right)$. Clearly, we only need to call the zeroth-order oracle twice per iteration. So the query complexity of Algorithm 1 is also $\mathcal{O}\left(\frac{\text{tr}(\mathbf{M})\sigma^2}{\lambda_{\min}^2(\mathbf{M})}\frac{1}{\varepsilon}\right)$. The iteration complexity of SGD is $\mathcal{O}\left(\frac{\lambda_{\max}(\mathbf{M})\sigma^2}{\lambda_{\min}^2(\mathbf{M})}\frac{1}{\varepsilon}\right)$ (Rakhlin et al., 2011). However, the number of times we call the zeroth-order oracles in each iteration is $2 \times d$. Then, the query complexity of ZSC is $\mathcal{O}\left(\frac{d\lambda_{\max}(\mathbf{M})\sigma^2}{\lambda_{\min}^2(\mathbf{M})}\frac{1}{\varepsilon}\right)$. So, ZSG is better than ZSC when the eigenvalues of the Hessian matrix are very different. That is to say, we only need to select the algorithm with better performance by comparing $\text{tr}(\mathbf{M})$ and $d\lambda_{\max}(\mathbf{M})$.

Then, we will generalize our results to the other functions based on Assumption 2.1. In other words, maybe $\gamma_l \neq 1$ or $\gamma_u \neq 1$.

**Theorem 4.7.** *If f is in the general form described in the problem (1) and Assumption 2.1,2.2 hold. Let $\boldsymbol{x}^{t+1}$ be updated according to Eq. (9). We choose a fixed step size*

$$\eta_t \equiv \eta \leq \frac{1}{12\gamma_u\text{tr}(\mathbf{M})}, \tag{18}$$

*where $\text{tr}(\mathbf{M}) = \max_{\boldsymbol{z}^t}\text{tr}(\mathbf{M}(\boldsymbol{z}^t))$, $\lambda_{\min}(\mathbf{M}) = \min_{\boldsymbol{z}^t}\lambda_{\min}(\mathbf{M}(\boldsymbol{z}^t))$ and $\lambda_{\max}(\mathbf{M}) = \max_{\boldsymbol{z}^t}\lambda_{\max}(\mathbf{M}(\boldsymbol{z}^t))$. We also define $P_2(\alpha^2) = \frac{[\lambda_{\max}(\mathbf{M})+2\gamma_u\eta](6+d)^3\gamma_u^2\alpha^2}{4\gamma_l\lambda_{\min}(\mathbf{M})}$. Then, we can obtain*

$$\mathbb{E}\left[f(\boldsymbol{x}^{t+1}) - f^*\right] \leq \frac{6\eta\gamma_u\text{tr}(\mathbf{M})\sigma^2}{\gamma_l\lambda_{\min}(\mathbf{M})} + P_2(\alpha^2) + \left[1 - \frac{1}{2}\eta\gamma_l\lambda_{\min}(\mathbf{M})\right]^t\left[f(x^0) - f^*\right]. $$

If $\sigma^2 = 0$ and a sufficiently small $\alpha$ is chosen, to find an $\varepsilon$-suboptimal solution, the iteration complexity is

$$t = \mathcal{O}\left(\frac{\gamma_u \text{tr}(\mathbf{M})}{\gamma_l \lambda_{\min}(\mathbf{M})} \log \frac{1}{\varepsilon}\right). \tag{19}$$

From Theorem 4.7, we can observe that ZSG may outperform the coordinate sketching version of the SEGA algorithm when $\text{tr}(\mathbf{M}) \ll d\lambda_{\max}(\mathbf{M})$ and $q = \mathcal{O}(1)$. This conclusion generalizes the results in (Wang et al., 2024).

**Theorem 4.8.** *If $f$ is in the general form described in the problem (1) and Assumption 2.1,2.2 hold. Let $\boldsymbol{x}^{t+1}$ be updated according to Eq. (9). Then, we let general $\eta_t = \frac{l}{\gamma+t}$ be decreasing and $\gamma > 0$. In addition, we assume that $t_{\max} = T$ and we can fix the intermediate parameter $l = \frac{3}{\gamma_l \lambda_{\min}(\mathbf{M})}$. We define $Q_2(\alpha^2) = \frac{[18+108\gamma_u\gamma_l\lambda_{\max}(\mathbf{M})\text{tr}(\mathbf{M})+3\gamma_l\lambda_{\max}(\mathbf{M})\lambda_{\min}(\mathbf{M})T](6+d)^3\gamma_u^2\alpha^2}{4\gamma_l^2\lambda_{\min}^2(\mathbf{M})}$. We choose initial step size*

$$\eta_0 = \frac{l}{\gamma} \leq \frac{1}{12\gamma_u \text{tr}(\mathbf{M})}, \tag{20}$$

*which means we can obtain a lower bound for parameter $\gamma$,*

$$\gamma \geq \frac{36\gamma_u \text{tr}(\mathbf{M})}{\lambda_{\min}(\mathbf{M})}. \tag{21}$$

*Then, we choose another parameter*

$$v = \max\left\{\gamma(f(\boldsymbol{x}^0) - f^*), \frac{54\gamma_u \text{tr}(\mathbf{M})\sigma^2}{\gamma_l^2\lambda_{\min}^2(\mathbf{M})} + Q_2(\alpha^2)\right\}, \tag{22}$$

*where $\text{tr}(\mathbf{M}) = \max_{\boldsymbol{z}^t} \text{tr}(\mathbf{M}(\boldsymbol{z}^t))$, $\lambda_{\min}(\mathbf{M}) = \min_{\boldsymbol{z}^t} \lambda_{\min}(\mathbf{M}(\boldsymbol{z}^t))$ and $\lambda_{\max}(\mathbf{M}) = \max_{\boldsymbol{z}^t} \lambda_{\max}(\mathbf{M}(\boldsymbol{z}^t))$. Then, we can obtain*

$$\mathbb{E}\left[f(\boldsymbol{x}^t) - f^*\right] \leq \frac{v}{\gamma+t}.$$

*If $\sigma^2 > 0$, to find an $\varepsilon$-suboptimal solution, the iteration complexity is*

$$t = \mathcal{O}\left(\left[\frac{\gamma_u \text{tr}(\mathbf{M})\sigma^2}{\gamma_l^2\lambda_{\min}(\mathbf{M})} + Q_2(\alpha^2)\right]\frac{1}{\varepsilon}\right). \tag{23}$$

From Theorem 4.8, if a sufficiently small $\alpha$ is chosen in practice, we can observe that ZSG may outperform ZSC when $\text{tr}(\mathbf{M}) \ll d\lambda_{\max}(\mathbf{M})$ and $\frac{q}{\gamma_l} = \mathcal{O}(1)$. For quadratic functions, we can easily find that $q = \frac{q}{\gamma_l} = 1$, which is consistent with our previous analysis. For other functions, we can impose further assumptions to show that the query complexity of algorithm ZSG is significantly improved compared to algorithm ZSC. This is an interesting direction to explore further. Without additional assumptions, improved global convergence for quadratic functions is the best result that can be hoped for.

## 5 EXPERIMENTS

We have provided a comprehensive theoretical analysis of ZSG in the preceding sections. This section is dedicated to the empirical validation of ZSG's effectiveness and superiority. We give the detailed structure of ZSC in Algorithm 2 which we intend to use for comparison.

### 5.1 QUADRATIC FUNCTIONS

In this part, our experiments will focus on the quadratic minimization problem, whose objective function adheres to the form delineated in the problem (1), characterized by

$$\min_{x\in\mathbb{R}^d} f(\boldsymbol{x}) = \frac{1}{2n}\boldsymbol{x}^\top \mathbf{A}\mathbf{A}^\top \boldsymbol{x} - \boldsymbol{b}^\top \boldsymbol{x}, \tag{24}$$

Table 1: Setting of diagonal matrix $\mathbf{\Sigma}$ used in Eq. (25) to construct $\mathbf{A}$.

| Type | $\mathbf{\Sigma}$ |
|---|---|
| 1 | $d = 100$ Matrix with first 99 components equal to 10 and the remaining one equal to $10\sqrt{10}$ |
| 2 | $d = 100$ Matrix with first 80 components equal to 10 and the rest equal to $10\sqrt{10}$ |
| 3 | $d = 500$ Matrix with first 499 components equal to $10\sqrt{5}$ and the remaining one equal to $100\sqrt{5}$ |
| 4 | $d = 500$ Matrix with first 480 components equal to $10\sqrt{5}$ and the rest equal to $100\sqrt{5}$ |

where $\mathbf{M} = \frac{1}{n}\mathbf{A}\mathbf{A}^\top$. The parameters of the quadratic function which we construct as follows: the dimension of feature vector $\boldsymbol{x}$ is $d$. We set

$$\mathbf{A} \stackrel{def}{=} \mathbf{U}\mathbf{\Sigma}\mathbf{U}^\top, \tag{25}$$

where $\mathbf{U}$ obtained from QR decomposition of random matrix with independent entries from $\mathcal{N}(0,1)$ and $\mathbf{\Sigma}$ is set as Table 1 and $\boldsymbol{b}$ is a random vector with independent entries drawn from $\mathcal{N}(0,1)$. For each problem, the starting point was chosen to be a vector with independent entries from $\mathcal{N}(0,1)$.

In this experiment, we compare ZSG with ZSC algorithm for problem described in Eq. (24). We properly choose the decreasing step sizes of them. According to the theoretical results of ZSG and ZSC, step sizes of these two algorithms should be proportional to $\mathcal{O}(1/(\mathrm{tr}(\mathbf{M}) + \lambda_{\min}(\mathbf{M})t))$ and $\mathcal{O}(1/(\lambda_{\min}(\mathbf{M})t))$, respectively. We report the experimental results in Figure 1.

When $d = 100$, we can observe that in the first two experiments, ZSG is faster than ZSC. As $\mathrm{tr}(\mathbf{M})$ increases, the running speed of ZSG slows down. As long as condition $\mathrm{tr}(\mathbf{M}) \ll d\lambda_{\max}(\mathbf{M})$ is met, ZSG is superior to ZSC.

When $d = 500$, we can observe that in the remaining two experiments, ZSG is significantly faster than ZSC. At the same time, we can observe similar results when $\mathrm{tr}(\mathbf{M})$ increases. It should be pointed out that as the dimension of the problem increases, the eigenvalues of Hessian matrix become more and more diverse, and ZSG is more likely to perform better than ZSC. All results match our theoretical analysis.

## 5.2 Logistic Regression for Binary Classification

In this part, we will use a real dataset to compare the convergence rates of ZSG and ZSC on the strongly convex function. We consider the logistic regression with a loss function

$$f(\boldsymbol{x}) = \frac{1}{n}\sum_{i=1}^{n} \log[1 + \exp(-y_i \langle a_i, \boldsymbol{x}\rangle)] + \frac{\beta}{2}\|\boldsymbol{x}\|^2,$$

where $a_i \in \mathbb{R}^d$ is the i-th input data, $y_i \in \{-1, 1\}$ is the corresponding label and $\beta$ is the regularizer parameter. We conduct experiments on 'mushrooms', 'phishing' and 'a8a' with $d = 112, 68$ and $d = 123$ respectively. These three datasets can be downloaded from libsvm datasets. The number of samples of 'mushrooms' is $n = 8124$ and the number of samples of 'phishing' is $n = 11055$. In our experiments on 'mushrooms' and 'phishing', we divide the training set and test set in a ratio of 4:1 and set $\beta = 0.001$. We properly choose the batch size $|\mathcal{S}|$ and the decreasing step sizes of them. We report the experimental results in Figure 2.

We report the training loss for all experiments in the three subgraphs of the first column. We can observe that ZSG achieve much faster convergence rate than ZSC. We report the test accuracy in the second column. We observe that the test accuracy of ZSG on the mushrooms dataset increases rapidly in the initial phase, exceeding that of ZSC. Subsequently, as the changes in test accuracy stabilize, ZSG's accuracy improves relative to ZSC. Furthermore, in the experiments on the phishing and a8a datasets, the test accuracy of ZSG exceeded that of ZSC. All in all, we can conclude that the convergence performance of the ZSG algorithm is superior to that of the ZSC algorithm in practice, while ZSG also achieves better test accuracy compared to ZSC. This result matches our theoretical analysis. A plain understanding is that ZSG can simultaneously handle all coordinates in a single oracle call, while ZSC processes one coordinate.

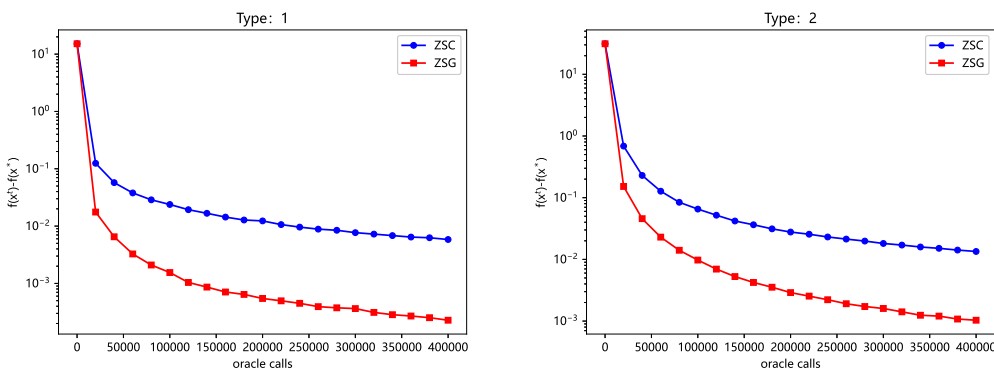

(a) The comparison on the first type diagonal matrix   (b) The comparison on the second type diagonal matrix

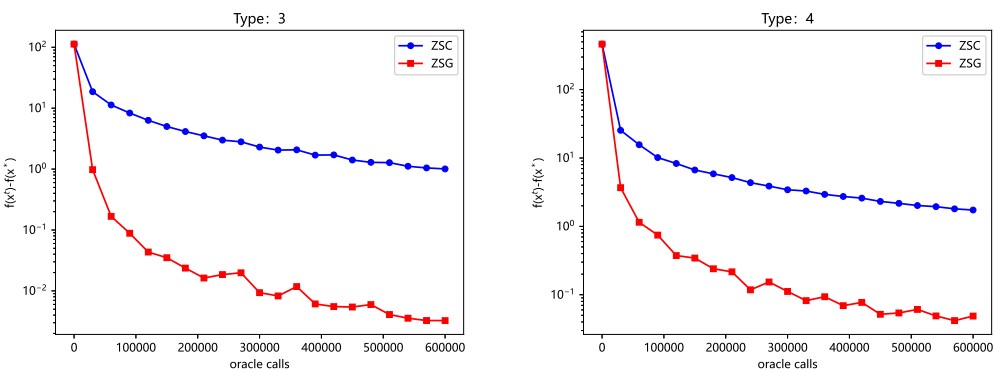

(c) The comparison on the third type diagonal matrix   (d) The comparison on the fourth type diagonal matrix

Figure 1: Comparison of running results of `ZSG` and `ZSC` on quadratic functions.

## 6   CONCLUSION AND FUTURE WORK

In this paper, we are the first to theoretically analyze the conditions under which the performance of algorithm `ZSG` exceeds that of algorithm `ZSC`. The most critical step is to verify whether condition $\operatorname{tr}(\mathbf{M}) \ll d\lambda_{\max}(\mathbf{M})$ holds. When the distribution of eigenvalues of the Hessian matrix varies significantly, the aforementioned condition can naturally hold. We obtain the best results for the analysis of quadratic functions. When $\sigma^2 = 0$, we get the main conclusion proposed by Wang et al. (2024): the complexity of `ZSG` is $\mathcal{O}\left(\frac{\operatorname{tr}(\mathbf{M})}{\lambda_{\min}(\mathbf{M})} \log \frac{1}{\varepsilon}\right)$ outperforms the coordinate sketching version of the `SEGA` algorithm whose complexity is $\mathcal{O}\left(\frac{d\lambda_{\max}(\mathbf{M})}{\lambda_{\min}(\mathbf{M})} \log \frac{1}{\varepsilon}\right)$ in the field of zeroth-order optimization. When $\sigma^2 > 0$, we obtain the main conclusion of our paper: the query complexity of `ZSG` is $\mathcal{O}\left(\frac{\operatorname{tr}(\mathbf{M})\sigma^2}{\lambda_{\min}^2(\mathbf{M})} \frac{1}{\varepsilon}\right)$, which outperforms `ZSC` algorithm, whose query complexity is $\mathcal{O}\left(\frac{d\lambda_{\max}(\mathbf{M})\sigma^2}{\lambda_{\min}^2(\mathbf{M})} \frac{1}{\varepsilon}\right)$. In other words, `ZSG` exhibits weak dimensional dependence. Both the synthetic datasets and the real dataset match our theoretical analysis. So, our research can contribute practical guidance in the field of zeroth-order optimization.

By retaining the upper and lower quadratic regularity constants $\gamma_u$ and $\gamma_l$, we extend our convergence analysis result from quadratic functions to any class of functions. We may need to consider more additional assumptions or conduct a more in-depth analysis to verify the conditions in the future under which $q = \mathcal{O}(1)$ or $\frac{q}{\gamma_l} = \mathcal{O}(1)$ holds.

In addition, a meaningful research direction is to incorporate information from the second-order Hessian matrix into the gradient estimation $\hat{\nabla} f(\boldsymbol{x}^t, \mathcal{S}_t)$. The motivation of this change come from a question: significant difference in curvature of loss function can lead to instability or decelerated

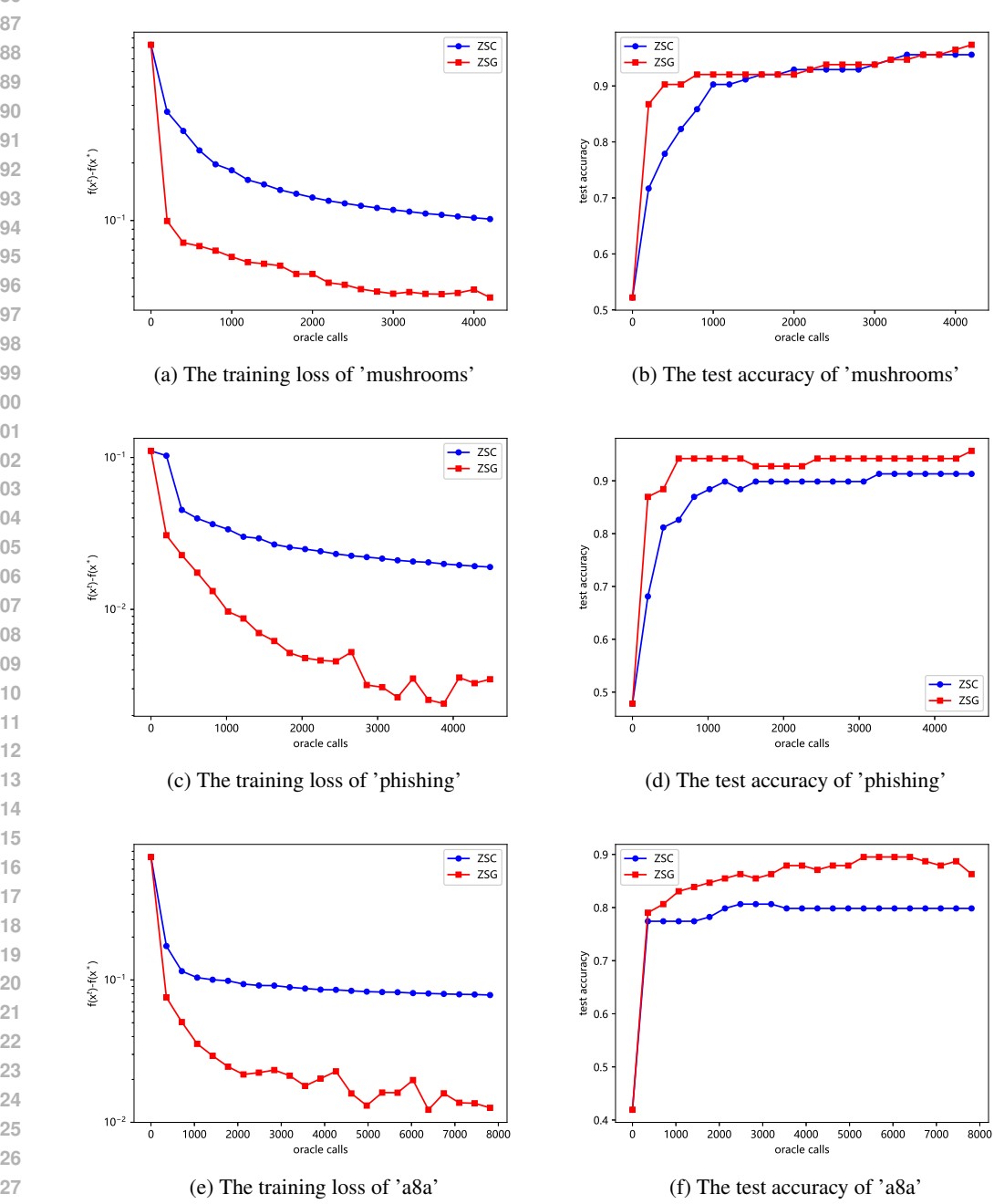

(a) The training loss of 'mushrooms'

(b) The test accuracy of 'mushrooms'

(c) The training loss of 'phishing'

(d) The test accuracy of 'phishing'

(e) The training loss of 'a8a'

(f) The test accuracy of 'a8a'

Figure 2: Comparison of running results of ZSG and ZSC on binary classification problem.

training. The Hessian information can be leveraged to effectively adjust the magnitude of the parameter updates solving the above dilemma. We believe that we can achieve better practical performance in terms of query complexity within our analytical framework, and then, extend the conclusion by using the quadratic regularity assumption.

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

---

**Algorithm 2** ZSC: ZO-SGD-Coordinate Method

---

**Input and Initialize:** parameters $\boldsymbol{x} \in \mathbb{R}^d$, loss function $f : \mathbb{R}^d \to \mathbb{R}$, step budget $t$, step size $\eta_t > 0$, perturbation scale $\alpha$, sample distribution $\mathcal{D}$, initial point $\boldsymbol{x}^0 \in \mathbb{R}^d$

**for** $t = 0, 1, \cdots$ **do**
  $\tilde{\nabla} f(\boldsymbol{x}^t, \mathcal{S}_t) = \mathbf{0}$
  Sample $\mathcal{S}_t \sim \mathcal{D}$
  **for** $i = 0, 1, \cdots, d$ **do**
    Query the zeroth-order oracle $f_+^t = f(\boldsymbol{x}^t + \alpha \boldsymbol{e}_i, \mathcal{S}_t)$
    Query the zeroth-order oracle $f_-^t = f(\boldsymbol{x}^t - \alpha \boldsymbol{e}_i, \mathcal{S}_t)$
    Estimating the partial derivative $\tilde{\nabla}_{\boldsymbol{e}_i} f(\boldsymbol{x}^t, \mathcal{S}_t) = \frac{(f_+^t - f_-^t)}{2\alpha} \cdot \boldsymbol{e}_i$
    $\tilde{\nabla} f(\boldsymbol{x}^t, \mathcal{S}_t) = \tilde{\nabla} f(\boldsymbol{x}^t, \mathcal{S}_t) + \tilde{\nabla}_{\boldsymbol{e}_i} f(\boldsymbol{x}^t, \mathcal{S}_t)$
  **end for**
  $\boldsymbol{x}^{t+1} = \boldsymbol{x}^t - \eta_t \tilde{\nabla} f(\boldsymbol{x}^t, \mathcal{S}_t)$
**end for**

---

# A    SEVERAL USEFUL LEMMAS

The following lemma shows that the expectation of the product of two quadratic forms of the random Gaussian vector is related to the trace of the corresponding matrix.

**Lemma A.1** (Magnus et al. (1978)). *Let $\mathbf{A}$ and $\mathbf{B}$ be two symmetric matrices, and $\boldsymbol{u}$ obeys the Gaussian distribution, that is, $\boldsymbol{u} \sim \mathcal{N}(\mathbf{0}, \mathbf{I}_d)$. Define $z = \boldsymbol{u}^\top \mathbf{A} \boldsymbol{u} \cdot \boldsymbol{u}^\top \mathbf{B} \boldsymbol{u}$. The expectation of $z$ is*

$$\mathbb{E}_{\boldsymbol{u}}[z] = (\text{tr}\mathbf{A})(\text{tr}\mathbf{B}) + 2(\text{tr}\mathbf{A}\mathbf{B}). \tag{26}$$

**Lemma A.2** (Nesterov & Spokoiny (2017)). *Let $\boldsymbol{u}$ obeys the Gaussian distribution, that is, $\boldsymbol{u} \sim \mathcal{N}(\mathbf{0}, \mathbf{I}_d)$. We define normalization constant $\kappa = \int e^{-\frac{1}{2}\|\boldsymbol{u}\|^2} d\boldsymbol{u}$ and define moments $\mathbf{M}_p = \frac{1}{\kappa} \int \|\boldsymbol{u}\|^p e^{-\frac{1}{2}\|\boldsymbol{u}\|^2} d\boldsymbol{u}$. For $p \geq 2$, we can obtain upper bounds*

$$n^{p/2} \leq \mathbf{M}_p \leq (p+d)^{p/2}. \tag{27}$$

**Lemma A.3.** *If we have a positive definite matrix $\mathbf{M}$ defined as weighted inner product, for all $\boldsymbol{x} \in \mathbb{R}^d$, we can obtain the following inequalities*

$$\|\boldsymbol{x}\|_{\mathbf{M}}^2 \leq \text{tr}(\mathbf{M}) \|\boldsymbol{x}\|^2, \tag{28}$$

$$\lambda_{\min}(\mathbf{M}) \|\boldsymbol{x}\|^2 \leq \|\boldsymbol{x}\|_{\mathbf{M}}^2 \leq \lambda_{\max}(\mathbf{M}) \|\boldsymbol{x}\|^2. \tag{29}$$

*Proof.* For a positive definite matrix $\mathbf{M}$, there must exist an orthogonal matrix $\mathbf{T}$ such that $\mathbf{M}$ is similar to a diagonal matrix whose elements are eigenvalues of matrix $\mathbf{M}$. We denote $\lambda_i$ be the i-th eigenvalue of matrix $\mathbf{M}$, then, we can obtain an equation as follows

$$\mathbf{M} = \mathbf{T} \text{diag}\{\lambda_1, \lambda_2, \cdots, \lambda_d\} \mathbf{T}^{-1}. \tag{30}$$

Let $\boldsymbol{y} = \mathbf{T}^\top \boldsymbol{x}$, then, we can easily prove this Lemma. We first prove Eq. (28)

$$\begin{aligned}
\|\boldsymbol{x}\|_{\mathbf{M}}^2 =& \langle \mathbf{M}\boldsymbol{x}, \boldsymbol{x} \rangle = \boldsymbol{x}^\top \mathbf{M}\boldsymbol{x} \overset{(30)}{=} x^\top \mathbf{T} \text{diag}\{\lambda_1, \lambda_2, \cdots, \lambda_d\} \mathbf{T}^{-1} x \\
=& \boldsymbol{x}^\top \mathbf{T} \text{diag}\{\lambda_1, \lambda_2, \cdots, \lambda_d\} \mathbf{T}^\top \boldsymbol{x} \\
=& \boldsymbol{y}^\top \text{diag}\{\lambda_1, \lambda_2, \cdots, \lambda_d\} \boldsymbol{y} \\
\leq& \text{tr}(\mathbf{M}) \boldsymbol{x}^\top \mathbf{T}\mathbf{T}^\top \boldsymbol{x} \\
=& \text{tr}(\mathbf{M}) \|\boldsymbol{x}\|^2.
\end{aligned}$$

Similarly, we can prove the Eq. (29). $\qquad\square$

**Lemma A.4.** *For the sake of simplicity in the subsequent proof, we first derive the upper bound of $\hat{\nabla} f(\boldsymbol{x}^t, \mathcal{S}_t)$. The upper bound is related to $\nabla f(\boldsymbol{x}^t, \mathcal{S}_t)$ and $\alpha$:*

$$\mathbb{E}_{\boldsymbol{u}_t}\left[ \|\hat{\nabla} f(\boldsymbol{x}^t, \mathcal{S}_t)\|_{\mathbf{M}(\boldsymbol{z}^t)}^2 \right] \leq 6\text{tr}(\mathbf{M}(\boldsymbol{z}^t)) \|\nabla f(\boldsymbol{x}^t, \mathcal{S}_t)\|^2 + \frac{(6+d)^3 \gamma_u^2 \alpha^2}{2}. \tag{31}$$

*Proof.* This part of the proof involves the basic properties of the norm and some important lemmas.

$$\mathbb{E}_{\boldsymbol{u}_t}\left[\|\hat{\nabla}f(\boldsymbol{x}^t,\mathcal{S}_t)\|^2_{\mathbf{M}(\boldsymbol{z}^t)}\right] \stackrel{(7)}{=} \mathbb{E}_{\boldsymbol{u}_t}\left[\left\|\boldsymbol{u}_t\boldsymbol{u}_t^\top\nabla f(\boldsymbol{x}^t,\mathcal{S}_t) + \phi(\boldsymbol{u}_t,\alpha,\boldsymbol{x}^t)\right\|^2_{\mathbf{M}(\boldsymbol{z}^t)}\right]$$

$$\leq 2\mathbb{E}_{\boldsymbol{u}_t}\left[\left\|\boldsymbol{u}_t\boldsymbol{u}_t^\top\nabla f(\boldsymbol{x}^t,\mathcal{S}_t)\right\|^2_{\mathbf{M}(\boldsymbol{z}^t)}\right] + 2\mathbb{E}_{\boldsymbol{u}_t}\left[\left\|\phi(\boldsymbol{u}_t,\alpha,\boldsymbol{x}^t)\right\|^2_{\mathbf{M}(\boldsymbol{z}^t)}\right]$$

$$\stackrel{(8)}{\leq} 2\mathbb{E}_{\boldsymbol{u}_t}\left[\left\|\boldsymbol{u}_t\boldsymbol{u}_t^\top\nabla f(\boldsymbol{x}^t,\mathcal{S}_t)\right\|^2_{\mathbf{M}(\boldsymbol{z}^t)}\right] + \frac{\gamma_u^2\alpha^2}{2}\mathbb{E}_{\boldsymbol{u}_t}\left[\|\boldsymbol{u}_t\|^6_{\mathbf{M}(\boldsymbol{z}^t)}\right]$$

$$\stackrel{(10)+(27)}{\leq} 6\mathrm{tr}(\mathbf{M}(\boldsymbol{z}^t))\left\|\nabla f(\boldsymbol{x}^t,\mathcal{S}_t)\right\|^2 + \frac{(6+d)^3\gamma_u^2\alpha^2}{2}.$$

$\square$

**Lemma A.5.** *For the sake of simplicity in the subsequent proof, we will derive the upper bound of an important inner product $\langle\nabla f(\boldsymbol{x}^t),\phi(\boldsymbol{u}_t,\alpha)\rangle$. The upper bound is related to real gradient $\nabla f(\boldsymbol{x}^t)$ and $\alpha$:*

$$-\mathbb{E}_{\boldsymbol{u}_t}\left[\langle\nabla f(\boldsymbol{x}^t),\phi(\boldsymbol{u}_t,\alpha,\boldsymbol{x}^t)\rangle\right] \leq \frac{1}{2}\left\|\nabla f(\boldsymbol{x}^t)\right\|^2 + \frac{\lambda_{\max}(\mathbf{M}(\boldsymbol{z}^t))(6+d)^3\gamma_u^2\alpha^2}{8}. \quad (32)$$

*Proof.* The techniques involved in this part are similar to those in Lemma A.4.

$$-\mathbb{E}_{\boldsymbol{u}_t}\left[\langle\nabla f(\boldsymbol{x}^t),\phi(\boldsymbol{u}_t,\alpha,\boldsymbol{x}^t)\rangle\right] \leq \mathbb{E}_{\boldsymbol{u}_t}\left[\left\|\nabla f(\boldsymbol{x}^t)\right\|\left\|\phi(\boldsymbol{u}_t,\alpha,\boldsymbol{x}^t)\right\|\right]$$

$$\leq \frac{1}{2}\left\|\nabla f(\boldsymbol{x}^t)\right\|^2 + \frac{1}{2}\mathbb{E}_{\boldsymbol{u}_t}\left[\left\|\phi(\boldsymbol{u}_t,\alpha,\boldsymbol{x}^t)\right\|^2\right]$$

$$\stackrel{(8)}{\leq} \frac{1}{2}\left\|\nabla f(\boldsymbol{x}^t)\right\|^2 + \frac{\gamma_u^2\alpha^2}{8}\mathbb{E}_{\boldsymbol{u}_t}\left[\|\boldsymbol{u}_t\|^4_{\mathbf{M}(\boldsymbol{z}^t)}\cdot\|\boldsymbol{u}_t\|^2\right]$$

$$\stackrel{(29)}{\leq} \frac{1}{2}\left\|\nabla f(\boldsymbol{x}^t)\right\|^2 + \frac{\lambda_{\max}(\mathbf{M}(\boldsymbol{z}^t))\gamma_u^2\alpha^2}{8}\mathbb{E}_{\boldsymbol{u}_t}\left[\|\boldsymbol{u}_t\|^6\right]$$

$$\stackrel{(27)}{\leq} \frac{1}{2}\left\|\nabla f(\boldsymbol{x}^t)\right\|^2 + \frac{\lambda_{\max}(\mathbf{M}(\boldsymbol{z}^t))(6+d)^3\gamma_u^2\alpha^2}{8}.$$

$\square$

# B PROOF OF IMPORTANT LEMMAS

In this section, we give some details of proof about some important Lemmas.

## B.1 PROOF OF LEMMA 3.2

*Proof.* By the Taylor's expansion, we can obtain that

$$f(\boldsymbol{x}+\alpha\boldsymbol{u},\mathcal{S}) = f(\boldsymbol{x}) + \alpha\langle\nabla f(\boldsymbol{x},\mathcal{S}),\boldsymbol{u}\rangle + \phi'(\boldsymbol{u},\alpha,\boldsymbol{x})$$

where $\phi'(\boldsymbol{u},\alpha,\boldsymbol{x}) = f(\boldsymbol{x}+\alpha\boldsymbol{u},\mathcal{S}) - f(\boldsymbol{x}) - \alpha\langle\nabla f(\boldsymbol{x},\mathcal{S}),\boldsymbol{u}\rangle$. Similarly, we can obtain

$$f(\boldsymbol{x}-\alpha\boldsymbol{u},\mathcal{S}) = f(\boldsymbol{x}) - \alpha\langle\nabla f(\boldsymbol{x},\mathcal{S}),\boldsymbol{u}\rangle + \phi'(\boldsymbol{u},-\alpha,\boldsymbol{x}).$$

$$\hat{\nabla}f(\boldsymbol{x},\mathcal{S}) = \frac{[f(\boldsymbol{x}+\alpha\boldsymbol{u},\mathcal{S}) - f(\boldsymbol{x}-\alpha\boldsymbol{u},\mathcal{S})]}{2\alpha}\cdot\boldsymbol{u} = \boldsymbol{u}\boldsymbol{u}^\top\nabla f(\boldsymbol{x},\mathcal{S}) + \frac{\phi'(\boldsymbol{u},\alpha,\boldsymbol{x}) - \phi'(\boldsymbol{u},-\alpha,\boldsymbol{x})}{2\alpha}\cdot\boldsymbol{u}.$$

By the upper quadratically regular assumption, we can obtain that

$$|\phi'(\boldsymbol{u},\alpha,\boldsymbol{x})| = |f(\boldsymbol{x}+\alpha\boldsymbol{u},\mathcal{S}) - f(\boldsymbol{x}) - \alpha\langle\nabla f(\boldsymbol{x},\mathcal{S}),\boldsymbol{u}\rangle| \leq \frac{\gamma_u\alpha^2}{2}\|\boldsymbol{u}\|^2_{\mathbf{M}(\boldsymbol{z}_1)},$$

$$|\phi'(\boldsymbol{u},-\alpha,\boldsymbol{x})| = |f(\boldsymbol{x}-\alpha\boldsymbol{u},\mathcal{S}) - f(\boldsymbol{x}) + \alpha\langle\nabla f(\boldsymbol{x},\mathcal{S}),\boldsymbol{u}\rangle| \leq \frac{\gamma_u\alpha^2}{2}\|\boldsymbol{u}\|^2_{\mathbf{M}(\boldsymbol{z}_2)}.$$

Then, we can finally obtain that

$$\left\|\frac{\phi'(\boldsymbol{u},\alpha,\boldsymbol{x}) - \phi'(\boldsymbol{u},-\alpha,\boldsymbol{x})}{2\alpha}\cdot\boldsymbol{u}\right\| \leq \frac{|\phi'(\boldsymbol{u},\alpha,\boldsymbol{x})| + |\phi'(\boldsymbol{u},-\alpha,\boldsymbol{x})|}{2\alpha}\|\boldsymbol{u}\| \leq \frac{\gamma_u\alpha}{2}\|\boldsymbol{u}\|^2_{\mathbf{M}(\boldsymbol{z})}\cdot\|\boldsymbol{u}\|.$$

$\square$

### B.2 PROOF OF LEMMA 4.1

*Proof.* This part of the proof mainly relies on the properties of the matrix trace.

$$
\begin{aligned}
\mathbb{E}_{\boldsymbol{u}_t}\left[\left\|\boldsymbol{u}_t\boldsymbol{u}_t^\top\nabla f(\boldsymbol{x}^t,\mathcal{S}_t)\right\|_{\mathbf{M}}^2\right] =&\mathbb{E}_{\boldsymbol{u}_t}\left[\nabla f(\boldsymbol{x}^t,\mathcal{S}_t)^\top\boldsymbol{u}_t\boldsymbol{u}_t^\top\mathbf{M}^\top\boldsymbol{u}_t\boldsymbol{u}_t^\top\nabla f(\boldsymbol{x}^t,\mathcal{S}_t)\right] \\
=&\mathbb{E}_{\boldsymbol{u}_t}\left[\mathrm{tr}(\nabla f(\boldsymbol{x}^t,\mathcal{S}_t)^\top\boldsymbol{u}_t\boldsymbol{u}_t^\top\mathbf{M}^\top\boldsymbol{u}_t\boldsymbol{u}_t^\top\nabla f(\boldsymbol{x}^t,\mathcal{S}_t))\right] \\
=&\mathbb{E}_{\boldsymbol{u}_t}\left[\mathrm{tr}(\boldsymbol{u}_t^\top\mathbf{M}^\top\boldsymbol{u}_t\boldsymbol{u}_t^\top\nabla f(\boldsymbol{x}^t,\mathcal{S}_t)\nabla f(\boldsymbol{x}^t,\mathcal{S}_t)^\top\boldsymbol{u}_t)\right] \\
\overset{(26)}{=}&\mathrm{tr}(\mathbf{M})\mathrm{tr}(\nabla f(\boldsymbol{x}^t,\mathcal{S}_t)\nabla f(\boldsymbol{x}^t,\mathcal{S}_t)^\top) \\
&+2\mathrm{tr}(\nabla f(\boldsymbol{x}^t,\mathcal{S}_t)^\top\mathbf{M}^\top\nabla f(\boldsymbol{x}^t,\mathcal{S}_t)) \\
=&\mathrm{tr}(\mathbf{M})\left\|\nabla f(\boldsymbol{x}^t,\mathcal{S}_t)\right\|^2+2\left\|\nabla f(\boldsymbol{x}^t,\mathcal{S}_t)\right\|_{\mathbf{M}}^2 \\
\overset{(28)}{\leq}&3\mathrm{tr}(\mathbf{M})\left\|\nabla f(\boldsymbol{x}^t,\mathcal{S}_t)\right\|^2.
\end{aligned}
$$

$\square$

### B.3 PROOF OF LEMMA 4.2

*Proof.* We use the lower quadratically regular introduced in Assumption 2.1,

$$
f(\boldsymbol{y}) \geq f(\boldsymbol{x}) + \langle\nabla f(\boldsymbol{x}),\boldsymbol{y}-\boldsymbol{x}\rangle + \frac{\gamma_l}{2}\left\|\boldsymbol{y}-\boldsymbol{x}\right\|_{\mathbf{M}(\boldsymbol{z})}^2.
$$

Then, we construct an auxiliary function,

$$
F(\boldsymbol{y}) = f(\boldsymbol{x}) + \langle\nabla f(\boldsymbol{x}),\boldsymbol{y}-\boldsymbol{x}\rangle + \frac{\gamma_l}{2}\left\|\boldsymbol{y}-\boldsymbol{x}\right\|_{\mathbf{M}(\boldsymbol{z})}^2.
$$

To obtain the minimum of the auxiliary function, we need to make

$$
\nabla F(\boldsymbol{y}^*) = \nabla f(\boldsymbol{x}) + 2\gamma_l\mathbf{M}(\boldsymbol{z})(\boldsymbol{y}^*-\boldsymbol{x}) = 0.
$$

So, we can find that

$$
\boldsymbol{y}^* = \boldsymbol{x} - \frac{1}{\gamma_l}\mathbf{M}(\boldsymbol{z})^{-1}\nabla f(\boldsymbol{x}). \tag{33}
$$

Using the above information, we can continue to deduce that

$$
\begin{aligned}
f(\boldsymbol{y}) \geq& F(\boldsymbol{y}) \\
\geq& F(\boldsymbol{y}^*) \\
\overset{(33)}{=}& f(\boldsymbol{x}) - \left\langle\nabla f(\boldsymbol{x}),\frac{1}{\gamma_l}\mathbf{M}(\boldsymbol{z})^{-1}\nabla f(\boldsymbol{x})\right\rangle + \frac{\gamma_l}{2}\left\|\frac{1}{\gamma_l}\mathbf{M}(\boldsymbol{z})^{-1}\nabla f(\boldsymbol{x})\right\|_{\mathbf{M}(\boldsymbol{z})}^2 \\
=& f(\boldsymbol{x}) - \frac{1}{\gamma_l}\left\|\nabla f(\boldsymbol{x})\right\|_{\mathbf{M}(\boldsymbol{z})^{-1}}^2 + \frac{1}{2\gamma_l}\nabla f(\boldsymbol{x})^\top(\mathbf{M}(\boldsymbol{z})^{-1})^\top\mathbf{M}(\boldsymbol{z})^\top\mathbf{M}(\boldsymbol{z})^{-1}\nabla f(\boldsymbol{x}) \\
=& f(\boldsymbol{x}) - \frac{1}{\gamma_l}\left\|\nabla f(\boldsymbol{x})\right\|_{\mathbf{M}(\boldsymbol{z})^{-1}}^2 + \frac{1}{2\gamma_l}\left\|\nabla f(\boldsymbol{x})\right\|_{\mathbf{M}(\boldsymbol{z})^{-1}}^2 \\
=& f(\boldsymbol{x}) - \frac{1}{2\gamma_l}\left\|\nabla f(\boldsymbol{x})\right\|_{\mathbf{M}(\boldsymbol{z})^{-1}}^2.
\end{aligned}
$$

Let $\boldsymbol{x} = \boldsymbol{x}^t, \boldsymbol{y} = \boldsymbol{x}^*$, and rearrange the above formula, we can obtain

$$
\begin{aligned}
f(\boldsymbol{x}^t) - f^* \leq& \frac{1}{2\gamma_l}\left\|\nabla f(\boldsymbol{x}^t)\right\|_{\mathbf{M}(\boldsymbol{z})^{-1}}^2 \\
\overset{(29)}{\leq}& \frac{\lambda_{\max}(\mathbf{M}(\boldsymbol{z})^{-1})}{2\gamma_l}\left\|\nabla f(\boldsymbol{x}^t)\right\| \\
=& \frac{1}{2\gamma_l\lambda_{\min}(\mathbf{M}(\boldsymbol{z}))}\left\|\nabla f(\boldsymbol{x}^t)\right\|.
\end{aligned}
$$

$\square$

## C PROOF OF MAIN THEOREMS

### C.1 PROOF OF THEOREM 4.3

*Proof.* Firstly, we can deduce the expectation of $f(\boldsymbol{x}^{t+1})$,

$$
\begin{aligned}
f(\boldsymbol{x}^{t+1}) &\overset{(2)}{\leq} f(\boldsymbol{x}^t) + \left\langle \nabla f(\boldsymbol{x}^t), \boldsymbol{x}^{t+1} - \boldsymbol{x}^t \right\rangle + \frac{1}{2} \left\| \boldsymbol{x}^{t+1} - \boldsymbol{x}^t \right\|_{\mathbf{M}}^2 \\
&\overset{(9)+(7)}{=} f(\boldsymbol{x}^t) - \eta \left\langle \nabla f(\boldsymbol{x}^t), \boldsymbol{u}_t \boldsymbol{u}_t^\top \nabla f(\boldsymbol{x}, \mathcal{S}) + \phi(\boldsymbol{u}_t, \alpha, \boldsymbol{x}^t) \right\rangle \\
&\quad + \frac{\eta^2}{2} \left\| \boldsymbol{u}_t \boldsymbol{u}_t^\top \nabla f(\boldsymbol{x}, \mathcal{S}) + \phi(\boldsymbol{u}_t, \alpha, \boldsymbol{x}^t) \right\|_{\mathbf{M}}^2 .
\end{aligned}
\tag{34}
$$

Let us deduce the expectation of $f(\boldsymbol{x}^{t+1})$ for $u$,

$$
\begin{aligned}
\mathbb{E}_{\boldsymbol{u}_t} \left[ f(\boldsymbol{x}^{t+1}) \right] &= f(\boldsymbol{x}^t) - \eta \left\langle \nabla f(\boldsymbol{x}^t), \mathbb{E}_{\boldsymbol{u}_t} \left[ \boldsymbol{u}_t \boldsymbol{u}_t^\top \nabla f(\boldsymbol{x}, \mathcal{S}) + \phi(\boldsymbol{u}_t, \alpha, \boldsymbol{x}^t) \right] \right\rangle \\
&\quad + \frac{\eta^2}{2} \mathbb{E}_{\boldsymbol{u}_t} \left[ \left\| \boldsymbol{u}_t \boldsymbol{u}_t^\top \nabla f(\boldsymbol{x}, \mathcal{S}) + \phi(\boldsymbol{u}_t, \alpha, \boldsymbol{x}^t) \right\|_{\mathbf{M}}^2 \right] \\
&\leq f(\boldsymbol{x}^t) - \eta \left\langle \nabla f(\boldsymbol{x}^t), \nabla f(\boldsymbol{x}^t, \mathcal{S}_t) \right\rangle - \eta \mathbb{E}_{\boldsymbol{u}_t} \left[ \left\langle \nabla f(\boldsymbol{x}^t), \phi(\boldsymbol{u}_t, \alpha, \boldsymbol{x}^t) \right\rangle \right] \\
&\quad + \frac{\eta^2}{2} \mathbb{E}_{\boldsymbol{u}_t} \left[ \left\| \boldsymbol{u}_t \boldsymbol{u}_t^\top \nabla f(\boldsymbol{x}, \mathcal{S}) + \phi(\boldsymbol{u}_t, \alpha, \boldsymbol{x}^t) \right\|_{\mathbf{M}}^2 \right] \\
&\overset{(31)+(32)}{\leq} f(\boldsymbol{x}^t) - \eta \left\langle \nabla f(\boldsymbol{x}^t), \nabla f(\boldsymbol{x}^t, \mathcal{S}_t) \right\rangle + \frac{\eta}{2} \left\| \nabla f(\boldsymbol{x}^t) \right\|^2 + 3\eta^2 \mathrm{tr}(\mathbf{M}) \left\| \nabla f(\boldsymbol{x}^t, \mathcal{S}_t) \right\|^2 \\
&\quad + \frac{\left[ \lambda_{\max}(\mathbf{M}) + 2\eta \right] (6+d)^3 \alpha^2 \eta}{8} .
\end{aligned}
$$

Then, let us deduce the expectation of $\mathbb{E}_u \left[ f(\boldsymbol{x}^{t+1}) \right]$,

$$
\begin{aligned}
\mathbb{E} \left[ f(\boldsymbol{x}^{t+1}) \right] &\leq f(\boldsymbol{x}^t) - \eta \left\langle \nabla f(\boldsymbol{x}^t), \mathbb{E} \left[ \nabla f(\boldsymbol{x}^t, \mathcal{S}_t) \right] \right\rangle + \frac{\eta}{2} \left\| \nabla f(\boldsymbol{x}^t) \right\|^2 \\
&\quad + 3\eta^2 \mathrm{tr}(\mathbf{M}) \mathbb{E} \left[ \left\| \nabla f(\boldsymbol{x}^t, \mathcal{S}_t) \right\|^2 \right] + \frac{\left[ \lambda_{\max}(\mathbf{M}) + 2\eta \right] (6+d)^3 \alpha^2 \eta}{8} \\
&= f(\boldsymbol{x}^t) - \frac{\eta}{2} \left\| \nabla f(\boldsymbol{x}^t) \right\|^2 + 3\eta^2 \mathrm{tr}(\mathbf{M}) \mathbb{E} \left[ \left\| \nabla f(\boldsymbol{x}^t, \mathcal{S}_t) \right\|^2 \right] \\
&\quad + \frac{\left[ \lambda_{\max}(\mathbf{M}) + 2\eta \right] (6+d)^3 \alpha^2 \eta}{8} \\
&\overset{(5)}{\leq} f(\boldsymbol{x}^t) - \frac{\eta}{2} \left\| \nabla f(\boldsymbol{x}^t) \right\|^2 + 3\eta^2 \mathrm{tr}(\mathbf{M})(\sigma^2 + \left\| \nabla f(\boldsymbol{x}^t) \right\|^2) \\
&\quad + \frac{\left[ \lambda_{\max}(\mathbf{M}) + 2\eta \right] (6+d)^3 \alpha^2 \eta}{8} \\
&= f(\boldsymbol{x}^t) + \left[ 3\eta^2 \mathrm{tr}(\mathbf{M}) - \frac{\eta}{2} \right] \left\| \nabla f(\boldsymbol{x}^t) \right\|^2 + 3\eta^2 \sigma^2 \mathrm{tr}(\mathbf{M}) \\
&\quad + \frac{\left[ \lambda_{\max}(\mathbf{M}) + 2\eta \right] (6+d)^3 \alpha^2 \eta}{8} \\
&= f(\boldsymbol{x}^t) + 3\eta^2 \sigma^2 \mathrm{tr}(\mathbf{M}) + \frac{\left[ \lambda_{\max}(\mathbf{M}) + 2\eta \right] (6+d)^3 \alpha^2 \eta}{8} \\
&\quad - \frac{\eta}{2} \left[ 1 - 6\eta \mathrm{tr}(\mathbf{M}) \right] \left\| \nabla f(\boldsymbol{x}^t) \right\|^2 \\
&\overset{(11)}{\leq} f(\boldsymbol{x}^t) + 3\eta^2 \sigma^2 \mathrm{tr}(\mathbf{M}) + \frac{\left[ \lambda_{\max}(\mathbf{M}) + 2\eta \right] (6+d)^3 \alpha^2 \eta}{8} \\
&\quad - \eta \lambda_{\min}(\mathbf{M}) \left[ 1 - 6\eta \mathrm{tr}(\mathbf{M}) \right] (f(\boldsymbol{x}^t) - f^*) \\
&\overset{(12)}{\leq} f(\boldsymbol{x}^t) + 3\eta^2 \sigma^2 \mathrm{tr}(\mathbf{M}) + \frac{\left[ \lambda_{\max}(\mathbf{M}) + 2\eta \right] (6+d)^3 \alpha^2 \eta}{8} \\
&\quad - \frac{1}{2} \eta \lambda_{\min}(\mathbf{M})(f(\boldsymbol{x}^t) - f^*) .
\end{aligned}
\tag{35}
$$

And then, let us use the optimal value $f^*$ to transform the inequality,

$$\mathbb{E}\left[f(\boldsymbol{x}^{t+1}) - f^*\right] + f^* - f(\boldsymbol{x}^t) \leq 3\eta^2\sigma^2\mathrm{tr}(\mathbf{M}) + \frac{\left[\lambda_{\max}(\mathbf{M}) + 2\eta\right](6+d)^3\alpha^2\eta}{8}$$

$$- \frac{1}{2}\eta\lambda_{\min}(\mathbf{M})\mathbb{E}\left[f(\boldsymbol{x}^t) - f^*\right].$$

Rearranging the above formula, we can obtain,

$$\mathbb{E}\left[f(\boldsymbol{x}^{t+1}) - f^*\right] \leq 3\eta^2\sigma^2\mathrm{tr}(\mathbf{M}) + \frac{\left[\lambda_{\max}(\mathbf{M}) + 2\eta\right](6+d)^3\alpha^2\eta}{8}$$

$$+ \left[1 - \frac{1}{2}\eta\lambda_{\min}(\mathbf{M})\right]\mathbb{E}\left[f(\boldsymbol{x}^t) - f^*\right].$$

We need to construct a recursive relation with the following structure,

$$\mathbb{E}\left[f(\boldsymbol{x}^{t+1}) - f^* - \beta\right] \leq \left[1 - \frac{1}{2}\eta\lambda_{\min}(\mathbf{M})\right]\mathbb{E}\left[f(x^t) - f^* - \beta\right].$$

If $\beta = \dfrac{24\eta\mathrm{tr}(\mathbf{M})\sigma^2 + \left[\lambda_{\max}(\mathbf{M}) + 2\eta\right](6+d)^3\alpha^2}{4\lambda_{\min}(\mathbf{M})}$, the above formula can be derived as

$$\mathbb{E}\left[f(\boldsymbol{x}^{t+1}) - f^* - \frac{24\eta\mathrm{tr}(\mathbf{M})\sigma^2 + \left[\lambda_{\max}(\mathbf{M}) + 2\eta\right](6+d)^3\alpha^2}{4\lambda_{\min}(\mathbf{M})}\right]$$

$$\leq \left[1 - \frac{1}{2}\eta\lambda_{\min}(\mathbf{M})\right]\mathbb{E}\left[f(x^t) - f^* - \frac{24\eta\mathrm{tr}(\mathbf{M})\sigma^2 + \left[\lambda_{\max}(\mathbf{M}) + 2\eta\right](6+d)^3\alpha^2}{4\lambda_{\min}(\mathbf{M})}\right]$$

$$\leq \left[1 - \frac{1}{2}\eta\lambda_{\min}(\mathbf{M})\right]^t\left[f(x^0) - f^* - \frac{24\eta\mathrm{tr}(\mathbf{M})\sigma^2 + \left[\lambda_{\max}(\mathbf{M}) + 2\eta\right](6+d)^3\alpha^2}{4\lambda_{\min}(\mathbf{M})}\right]$$

$$\leq \left[1 - \frac{1}{2}\eta\lambda_{\min}(\mathbf{M})\right]^t\left[f(x^0) - f^*\right].$$

Thus, we can obtain that

$$\mathbb{E}\left[f(\boldsymbol{x}^{t+1}) - f^*\right] \leq \frac{24\eta\mathrm{tr}(\mathbf{M})\sigma^2 + \left[\lambda_{\max}(\mathbf{M}) + 2\eta\right](6+d)^3\alpha^2}{4\lambda_{\min}(\mathbf{M})}$$

$$+ \left[1 - \frac{1}{2}\eta\lambda_{\min}(\mathbf{M})\right]^t\left[f(x^0) - f^*\right].$$

$\square$

### C.2 PROOF OF THEOREM 4.5

*Proof.* Firstly, if we choose decreasing step size $\eta_t$, based on C.1, we can obtain the following formula

$$\mathbb{E}\left[f(\boldsymbol{x}^{t+1}) - f(\boldsymbol{x}^t)\right] \leq 3\eta_t^2\sigma^2\mathrm{tr}(\mathbf{M}) + \frac{\left[\lambda_{\max}(\mathbf{M}) + 2\eta_t\right](6+d)^3\alpha^2\eta_t}{8}$$

$$- \eta_t\lambda_{\min}(\mathbf{M})\left[1 - 6\eta_t\mathrm{tr}(\mathbf{M})\right]\mathbb{E}\left[f(\boldsymbol{x}^t) - f^*\right]$$

$$\leq 3\eta_t^2\sigma^2\mathrm{tr}(\mathbf{M}) + \frac{\left[\lambda_{\max}(\mathbf{M}) + 2\eta_t\right](6+d)^3\alpha^2\eta_t}{8}$$

$$- \eta_t\lambda_{\min}(\mathbf{M})\left[1 - 6\eta_0\mathrm{tr}(\mathbf{M})\right]\mathbb{E}\left[f(\boldsymbol{x}^t) - f^*\right]$$

$$\overset{(14)}{\leq} 3\eta_t^2\sigma^2\mathrm{tr}(\mathbf{M}) + \frac{\left[\lambda_{\max}(\mathbf{M}) + 2\eta_t\right](6+d)^3\alpha^2\eta_t}{8}$$

$$- \frac{1}{2}\eta_t\lambda_{\min}(\mathbf{M})\mathbb{E}\left[f(\boldsymbol{x}^t) - f^*\right].$$

Let us prove the final result by induction, for $t = 0$

$$\mathbb{E}\left[f(\boldsymbol{x}^0) - f^*\right] = f(\boldsymbol{x}^0) - f^* = \frac{\gamma}{\gamma+0}\left[f(\boldsymbol{x}^0) - f^*\right] \leq \frac{v}{\gamma+0},$$

by the definition of $v$.

Suppose that holds for $t > 0$, then

$$\mathbb{E}\left[f(\boldsymbol{x}^{t+1}) - f^*\right] \leq 3\eta_t^2\sigma^2\text{tr}(\mathbf{M}) + \frac{\left[\lambda_{\max}(\mathbf{M}) + 2\eta_t\right](6+d)^3\alpha^2\eta_t}{8}$$

$$+ \left[1 - \frac{1}{2}\eta_t\lambda_{\min}(\mathbf{M})\right]\mathbb{E}\left[f(\boldsymbol{x}^t) - f^*\right]$$

$$\leq 3\eta_t^2\sigma^2\text{tr}(\mathbf{M}) + \frac{\left[\lambda_{\max}(\mathbf{M}) + 2\eta_t\right](6+d)^3\alpha^2\eta_t}{8}$$

$$+ \left[1 - \frac{1}{2}\eta_t\lambda_{\min}(\mathbf{M})\right]\frac{v}{\gamma + t}$$

$$= \frac{3\sigma^2l^2\text{tr}(\mathbf{M})}{(\gamma+t)^2} + \frac{\lambda_{\max}(\mathbf{M})(6+d)^3\alpha^2l}{8(\gamma+t)} + \frac{(6+d)^3\alpha^2l^2}{4(\gamma+t)^2}$$

$$+ \left[1 - \frac{l\lambda_{\min}(\mathbf{M})}{2(\gamma+t)}\right]\frac{v}{\gamma+t}$$

$$= \frac{(\gamma+t-1)v}{(\gamma+t)^2} + \frac{3\sigma^2l^2\text{tr}(\mathbf{M})}{(\gamma+t)^2} + \frac{\lambda_{\max}(\mathbf{M})(6+d)^3\alpha^2l}{8(\gamma+t)} + \frac{(6+d)^3\alpha^2l^2}{4(\gamma+t)^2}$$

$$- \frac{(l\lambda_{\min}(\mathbf{M})-2)v}{2(\gamma+t)^2}.$$

We let $\dfrac{3\sigma^2l^2\text{tr}(\mathbf{M})}{(\gamma+t)^2} + \dfrac{\lambda_{\max}(\mathbf{M})(6+d)^3\alpha^2l}{8(\gamma+t)} + \dfrac{(6+d)^3\alpha^2l^2}{4(\gamma+t)^2} - \dfrac{(l\lambda_{\min}(\mathbf{M})-2)v}{2(\gamma+t)^2} \leq 0.$ This is equivalent to

$$6\sigma^2l^2\text{tr}(\mathbf{M}) + \frac{(6+d)^3\alpha^2l^2}{2} + \frac{\lambda_{\max}(\mathbf{M})(6+d)^3\alpha^2l(\gamma+t)}{4} \leq (l\lambda_{\min}(\mathbf{M})-2)v.$$

$$\Rightarrow v \geq \frac{54\text{tr}(\mathbf{M})\sigma^2}{\lambda_{\min}^2(\mathbf{M})} + \frac{9(6+d)^3\alpha^2}{2\lambda_{\min}^2(\mathbf{M})} + \frac{3\lambda_{\max}(\mathbf{M})(6+d)^3\alpha^2(\gamma+t)}{4\lambda_{\min}(\mathbf{M})}.$$

$$\Rightarrow v \geq \frac{54\text{tr}(\mathbf{M})\sigma^2}{\lambda_{\min}^2(\mathbf{M})} + \frac{9(6+d)^3\alpha^2}{2\lambda_{\min}^2(\mathbf{M})} + \frac{3\lambda_{\max}(\mathbf{M})(6+d)^3\alpha^2(\gamma+T)}{4\lambda_{\min}(\mathbf{M})}$$

$$\overset{(15)}{\geq} \frac{54\text{tr}(\mathbf{M})\sigma^2}{\lambda_{\min}^2(\mathbf{M})} + \frac{\left[18 + 108\lambda_{\max}(\mathbf{M})\text{tr}(\mathbf{M}) + 3\lambda_{\max}(\mathbf{M})\lambda_{\min}(\mathbf{M})T\right](6+d)^3\alpha^2}{4\lambda_{\min}^2(\mathbf{M})}$$

$$= \frac{54\text{tr}(\mathbf{M})\sigma^2}{\lambda_{\min}^2(\mathbf{M})} + Q_1(\alpha^2).$$

So, we can finally obtain $v \geq \dfrac{54\text{tr}(\mathbf{M})\sigma^2}{\lambda_{\min}^2(\mathbf{M})} + Q_1(\alpha^2).$

Due to the facts

$$(\gamma+t)^2 \geq (\gamma+t+1)(\gamma+t-1) = (\gamma+t)^2 - 1,$$

then

$$\mathbb{E}\left[f(\boldsymbol{x}^{t+1}) - f^*\right] \leq \frac{v}{\gamma+t+1}.$$

$\square$

### C.3 PROOF OF THEOREM 4.7

If the objective function is not quadratic function, we notice that $\gamma_u \neq 1$ and $\gamma_l \neq 1$. So, we can transform inequality (34) into

$$f(\boldsymbol{x}^{t+1}) \leq f(\boldsymbol{x}^t) - \eta\left\langle\nabla f(\boldsymbol{x}^t), \boldsymbol{u}_t\boldsymbol{u}_t^\top\nabla f(\boldsymbol{x}^t, \mathcal{S}_t) + \phi(\boldsymbol{u}_t, \alpha, \boldsymbol{x}^t)\right\rangle$$

$$+ \frac{\gamma_u\eta^2}{2}\left\|\boldsymbol{u}_t\boldsymbol{u}_t^\top\nabla f(\boldsymbol{x}^t, \mathcal{S}_t) + \phi(\boldsymbol{u}_t, \alpha, \boldsymbol{x}^t)\right\|_{\mathbf{M}(\boldsymbol{z}^t)}^2.$$

Let us deduce the expectation of $f(\boldsymbol{x}^{t+1})$ for $\boldsymbol{u}$,

$$\mathbb{E}_{\boldsymbol{u}_t}\left[f(\boldsymbol{x}^{t+1})\right] \overset{(31)+(32)}{\leq} f(\boldsymbol{x}^t) + 3\eta^2\gamma_u\mathrm{tr}(\mathbf{M}(\boldsymbol{z}^t))\left\|\nabla f(\boldsymbol{x}^t, \mathcal{S}_t)\right\|^2 + \frac{\eta}{2}\left\|\nabla f(\boldsymbol{x}^t)\right\|^2$$
$$- \eta\left\langle \nabla f(\boldsymbol{x}^t), \nabla f(\boldsymbol{x}^t, \mathcal{S}_t)\right\rangle + \frac{[\lambda_{\max}(\mathbf{M}(\boldsymbol{z}^t)) + 2\gamma_u\eta](6+d)^3\gamma_u^2\alpha^2\eta}{8}.$$

And we can transform inequality (35) into

$$\mathbb{E}\left[f(\boldsymbol{x}^{t+1})\right] \leq f(\boldsymbol{x}^t) + 3\eta^2\sigma^2\gamma_u\mathrm{tr}(\mathbf{M}(\boldsymbol{z}^t)) + \frac{[\lambda_{\max}(\mathbf{M}(\boldsymbol{z}^t)) + 2\gamma_u\eta](6+d)^3\gamma_u^2\alpha^2\eta}{8}$$
$$- \frac{1}{2}\gamma_l\eta\lambda_{\min}(\mathbf{M}(\boldsymbol{z}^t))(f(\boldsymbol{x}^t) - f^*).$$

If we let $\mathrm{tr}(\mathbf{M}) = \max_{\boldsymbol{z}^t}\mathrm{tr}(\mathbf{M}(\boldsymbol{z}^t))$, $\lambda_{\min}(\mathbf{M}) = \min_{\boldsymbol{z}^t}\lambda_{\min}(\mathbf{M}(\boldsymbol{z}^t))$ and $\lambda_{\max}(\mathbf{M}) = \max_{\boldsymbol{z}^t}\lambda_{\max}(\mathbf{M}(\boldsymbol{z}^t))$ in the subsequent analysis. Then, we can obtain

$$\mathbb{E}\left[f(\boldsymbol{x}^{t+1}) - f^* - \frac{24\eta\gamma_u\mathrm{tr}(\mathbf{M})\sigma^2 + [\lambda_{\max}(\mathbf{M}) + 2\gamma_u\eta](6+d)^3\gamma_u^2\alpha^2}{4\gamma_l\lambda_{\min}(\mathbf{M})}\right]$$
$$\leq \left[1 - \frac{1}{2}\eta\gamma_l\lambda_{\min}(\mathbf{M})\right]\mathbb{E}\left[f(x^t) - f^* - \frac{24\eta\gamma_u\mathrm{tr}(\mathbf{M})\sigma^2 + [\lambda_{\max}(\mathbf{M}) + 2\gamma_u\eta](6+d)^3\gamma_u^2\alpha^2}{4\gamma_l\lambda_{\min}(\mathbf{M})}\right]$$
$$\leq \left[1 - \frac{1}{2}\eta\gamma_l\lambda_{\min}(\mathbf{M})\right]^t\left[f(x^0) - f^* - \frac{24\eta\gamma_u\mathrm{tr}(\mathbf{M})\sigma^2 + [\lambda_{\max}(\mathbf{M}) + 2\gamma_u\eta](6+d)^3\gamma_u^2\alpha^2}{4\gamma_l\lambda_{\min}(\mathbf{M})}\right]$$
$$\leq \left[1 - \frac{1}{2}\eta\gamma_l\lambda_{\min}(\mathbf{M})\right]^t\left[f(x^0) - f^*\right].$$

Thus, we can obtain that

$$\mathbb{E}\left[f(\boldsymbol{x}^{t+1}) - f^*\right] \leq \frac{24\eta\gamma_u\mathrm{tr}(\mathbf{M})\sigma^2 + [\lambda_{\max}(\mathbf{M}) + 2\gamma_u\eta](6+d)^3\gamma_u^2\alpha^2}{4\gamma_l\lambda_{\min}(\mathbf{M})}$$
$$+ \left[1 - \frac{1}{2}\eta\gamma_l\lambda_{\min}(\mathbf{M})\right]^t\left[f(x^0) - f^*\right].$$

Let $\sigma = 0$ and a sufficiently small $\alpha$ is chosen, similar to the proof process of D.1, we can obtain the iteration complexity

$$t = \mathcal{O}\left(\frac{\gamma_u\mathrm{tr}(\mathbf{M})}{\gamma_l\lambda_{\min}(\mathbf{M})}\log\frac{1}{\varepsilon}\right). \tag{36}$$

### C.4 PROOF OF THEOREM 4.8

Firstly, if we choose decreasing step size $\eta_t$, we can obtain the following formula

$$\mathbb{E}\left[f(\boldsymbol{x}^{t+1})\right] \leq f(\boldsymbol{x}^t) + 3\eta_t^2\sigma^2\gamma_u\mathrm{tr}(\mathbf{M}(\boldsymbol{z}^t)) + \frac{[\lambda_{\max}(\mathbf{M}(\boldsymbol{z}^t)) + 2\gamma_u\eta_t](6+d)^3\gamma_u^2\alpha^2\eta_t}{8}$$
$$- \frac{1}{2}\gamma_l\eta_t\lambda_{\min}(\mathbf{M}(\boldsymbol{z}^t))(f(\boldsymbol{x}^t) - f^*).$$

We let $\mathrm{tr}(\mathbf{M}) = \max_{\boldsymbol{z}^t}\mathrm{tr}(\mathbf{M}(\boldsymbol{z}^t))$, $\lambda_{\min}(\mathbf{M}) = \min_{\boldsymbol{z}^t}\lambda_{\min}(\mathbf{M}(\boldsymbol{z}^t))$ and $\lambda_{\max}(\mathbf{M}) = \max_{\boldsymbol{z}^t}\lambda_{\max}(\mathbf{M}(\boldsymbol{z}^t))$ in the subsequent analysis. Then, we need to add $\gamma_u$ and $\gamma_l$ to the appropriate position in the proof process of C.2 like the similar ways we operated in C.3. Suppose that

holds for $t > 0$, then

$$
\begin{aligned}
\mathbb{E}\left[f(\boldsymbol{x}^{t+1}) - f^*\right] \leq & 3\eta_t^2\sigma^2\gamma_u\mathrm{tr}(\mathbf{M}) + \frac{\left[\lambda_{\max}(\mathbf{M}) + 2\gamma_u\eta_t\right](6+d)^3\gamma_u^2\alpha^2\eta_t}{8} \\
& + \left[1 - \frac{1}{2}\gamma_l\eta_t\lambda_{\min}(\mathbf{M})\right]\mathbb{E}\left[f(\boldsymbol{x}^t) - f^*\right] \\
\leq & 3\eta_t^2\sigma^2\gamma_u\mathrm{tr}(\mathbf{M}) + \frac{\left[\lambda_{\max}(\mathbf{M}) + 2\gamma_u\eta_t\right](6+d)^3\gamma_u^2\alpha^2\eta_t}{8} \\
& + \left[1 - \frac{1}{2}\gamma_l\eta_t\lambda_{\min}(\mathbf{M})\right]\frac{v}{\gamma + t} \\
= & \frac{(\gamma + t - 1)v}{(\gamma + t)^2} + \frac{3\sigma^2l^2\gamma_u\mathrm{tr}(\mathbf{M})}{(\gamma + t)^2} + \frac{\lambda_{\max}(\mathbf{M})(6+d)^3\gamma_u^2\alpha^2l}{8(\gamma + t)} \\
& + \frac{(6+d)^3\gamma_u^3\alpha^2l^2}{4(\gamma + t)^2} - \frac{(\gamma_l l\lambda_{\min}(\mathbf{M}) - 2)v}{2(\gamma + t)^2}.
\end{aligned}
$$

We define $Q_2(\alpha^2) = \dfrac{\left[18 + 108\gamma_u\gamma_l\lambda_{\max}(\mathbf{M})\mathrm{tr}(\mathbf{M}) + 3\gamma_l\lambda_{\max}(\mathbf{M})\lambda_{\min}(\mathbf{M})T\right](6+d)^3\gamma_u^2\alpha^2}{4\gamma_l^2\lambda_{\min}^2(\mathbf{M})}.$

Then, we let $\dfrac{3\sigma^2l^2\gamma_u\mathrm{tr}(\mathbf{M})}{(\gamma + t)^2} + \dfrac{\lambda_{\max}(\mathbf{M})(6+d)^3\gamma_u^2\alpha^2l}{8(\gamma + t)} + \dfrac{(6+d)^3\gamma_u^3\alpha^2l^2}{4(\gamma + t)^2} - \dfrac{(\gamma_l l\lambda_{\min}(\mathbf{M}) - 2)v}{2(\gamma + t)^2} \leq$

$0$, which is equivalent to $v \geq \dfrac{54\gamma_u\mathrm{tr}(\mathbf{M})\sigma^2}{\gamma_l^2\lambda_{\min}(\mathbf{M})} + Q_2(\alpha^2).$

Finally, we obtain the iteration complexity

$$
t = \mathcal{O}\left(\left[\frac{\gamma_u\mathrm{tr}(\mathbf{M})\sigma^2}{\gamma_l^2\lambda_{\min}(\mathbf{M})} + Q_2(\alpha^2)\right]\frac{1}{\varepsilon}\right). \tag{37}
$$

# D  PROOF OF MAIN COROLLARIES

## D.1  PROOF OF COROLLARY 4.4

*Proof.* From the proof of Theorem 4.3, if we choose a sufficiently small $\alpha$ in practice, we can find that

$$
\begin{aligned}
\mathbb{E}\left[f(\boldsymbol{x}^{t+1}) - f^*\right] \leq & \frac{6\eta\mathrm{tr}(\mathbf{M})\sigma^2}{\lambda_{\min}(\mathbf{M})} + \left[1 - \frac{1}{2}\eta\lambda_{\min}(\mathbf{M})\right]^t\left[f(x^0) - f^*\right] \\
\overset{\sigma=0}{=} & \left[1 - \frac{1}{2}\eta\lambda_{\min}(\mathbf{M})\right]^t\left[f(\boldsymbol{x}^0) - f^*\right] \\
\leq & \exp\left(-\frac{1}{2}\eta\lambda_{\min}(\mathbf{M})t\right)\left[f(\boldsymbol{x}^0) - f^*\right] \\
\overset{(12)}{\leq} & \exp\left(-\frac{\lambda_{\min}(\mathbf{M})}{24\mathrm{tr}(\mathbf{M})}t\right)\left[f(\boldsymbol{x}^0) - f^*\right].
\end{aligned}
$$

Thus, in order to achieve $\varepsilon$-suboptimal solution, $t$ is required to be

$$
\begin{aligned}
t = & \frac{24\mathrm{tr}(\mathbf{M})}{\lambda_{\min}(\mathbf{M})}\left(\log\frac{1}{\varepsilon} + \log\left(f(\boldsymbol{x}^0) - f^*\right)\right) \\
= & \mathcal{O}\left(\frac{\mathrm{tr}(\mathbf{M})}{\lambda_{\min}(\mathbf{M})}\log\frac{1}{\varepsilon}\right).
\end{aligned}
$$

$\square$

