# OpenReview forum: "The Advancement in Stochastic Zeroth-Order Optimization: Mechanism of Accelerated Convergence of Gaussian Direction on Objectives with Skewed Hessian Eigenvalues"
_ICLR.cc/2025/Conference — Submitted to ICLR 2025_

### Official Review · Reviewer_rsNb · 2024-10-31

**Soundness:** 3
**Presentation:** 3
**Contribution:** 2
**Rating:** 5
**Confidence:** 3

**Summary:**

The paper studies zeroth-order optimization methods (aka. black-box optimization). Specifically, the authors provide theoretical explanation for an observation in practice that the zeroth-order Gaussian gradient descent (ZSG) usually outperforms the zeroth-order version of coordinate descent (ZSC). They show that the improvement mainly comes from the skewness of the Hessian matrix of the objective function. Loosely speaking, the iteration complexity of ZSG scales with $Trace(H)/ \lambda_{\min}(H)$, while the complexity of ZSC scales with $d*\lambda_{\max}(H)/\lambda_{\min}(H)$ where $H$ is the Hessian matrix, $d$ is the dimension. They also perform numerical experiments to show that ZSG outpeforms ZSC in practice.

**Strengths:**

The paper is well-written and the research motivation is clear. Zeroth-order optimization plays a crucial role in domains where privacy in training is essential or where model size makes gradient computation impractical. The theoretical findings are also interesting.

**Weaknesses:**

While the authors managed to show the improvement of ZSG over ZSC when the objective's Hessian is skewed for the quadratic function, the claim is not so clear for the general function. For example, the factor $\gamma_u / \gamma_l^2$ in eq (23) can be quite uncontrollable compared to the advantage gained from the skewness. I suggest the authors discuss this trade-off more, i.e., pinpoint cases where the improvement is meaningful.

**Questions:**

(1) Does ZSC have the "alpha term" that you ignored from the complexity of ZSG?  Does this alpha term affect the comparison between the two algorithms given that it is not ignored?

(2) Please provide more explanation on the sentence in lines 172 and 173. E.g., why are those parameters independent of the condition number of the data?

(3) Regarding conditions (2) and (3), is it  "exists z" or "for all z"? Please discuss these conditions more, especially how strong they are compared to "L-smooth" and "strongly convex"? Some discussions would be helpful instead of just citing [Frangella el at. 2023].

---

> ### Author Response · Authors · 2024-11-30
>
> Dear Reviewer rsNb,
>
> Thank you very much for your feedback. We will address each of the concerns and issues you raised, and provide clarifications accordingly.
>
> 1.Quadratic regularity and the quadratic regularity ratio generalize the notions of strong convexity, smoothness, and condition number to the Hessian norm. It is important to note that our main proof focuses on quadratic functions. The upper and lower quadratic regularization constants help us generalize the results, although the factor of $\frac{\gamma_u}{\gamma_l^2}$ is indeed difficult to control. Frangella et al. (2023) point out that for any objective with a Lipschitz Hessian, the quadratic regularity ratio approaches one as the optimal value is approached. Many studies also consider proving convergence rates locally. Additionally, our experiments not only include quadratic objectives but also logistic regression. From the experimental perspective, we further validate the advantages of the ZSG algorithm on other objective functions.
>
> 2.There exists a term involving $\alpha$, but it does not affect the comparison between the two algorithms. The impact of the $\alpha$-related term on the iterative complexity is negligible, as in practice, $\alpha$ can be chosen sufficiently small to effectively eliminate its influence.
>
> 3.Frangella et al. (2023) prove that under the quadratic regularization assumption, the convergence rate of algorithms such as SketchySVRG is independent of the condition number, and that quadratic regularization provides a tighter bound than condition number-based methods. Because the quadratic regularity ratio equals one for quadratic objectives and approaches one as the iterate approaches the optimum for any objective with a Lipschitz Hessian.
>
> 4.It refers to the fact that conditions (2) and (3) hold for all z. Notably, the bounds guaranteed by smoothness and strong convexity are looser than the bounds guaranteed by the quadratic regularity assumption. Frangella et al. (2023) prove that if F is L-smooth and µ-strongly convex, then, F is quadratically regular with $\frac{\mu}{L}  \le \gamma_l \le \gamma_u \le \frac{L}{\mu}$. To see why, suppose F is an ill-conditioned quadratic. Clearly, $\gamma_l=\gamma_u=1$ and $\frac{\mu}{L}  \le 1 \le \frac{L}{\mu}$. In addition, the quadratic regularity and the quadratic regularity ratio generalize the notions of strong convexity, smoothness, and condition number to the Hessian norm. We also suppose F is an ill-conditioned quadratic. In the case of the L-smoothness assumption, the quadratic term is weighted by a matrix whose diagonal entries correspond to the largest eigenvalue of the Hessian matrix. In contrast, for the quadratic regularization assumption, the quadratic term is simply weighted by the Hessian matrix itself. The latter is a weaker condition and provides a tighter upper bound.
>
> We would like to express our sincere gratitude once again for your valuable feedback！

---

> > ### Comment · Reviewer_rsNb · 2024-12-02
> >
> > Thank you for your reply. I appreciate the difficulty when working with general functions (with quadratic regularity assumption). However, if the analysis mainly focuses on quadratic function -- we are not able to claim improvement in the general case, I do not think it is significant enough and am not able to raise the score.
> >
> > Future suggestion: I believe the gain ratio $\frac{Tr(M)}{\lambda_{min}(M)}$ is closely related to the extra term $\frac{\gamma_u}{\gamma_l^2}$ as both of them are problem's parameters. If the authors manage to show that the trade-off is meaningful, e.g., the gain is not dominated by the extra term, the improvement is indeed significant.

---

### Official Review · Reviewer_cWQZ · 2024-11-01

**Soundness:** 1
**Presentation:** 1
**Contribution:** 1
**Rating:** 3
**Confidence:** 5

**Summary:**

The paper studies zeroth-order methods for finite-sum optimization and compares the complexity of two algorithms, ZSG and ZSC. The authors claim in the paper to rigorously and theoretically prove that ZSG is better than ZSC, under the quadratic regularity assumption.

**Strengths:**

The study on zeroth-order optimization is a trendy and important topic in optimization and machine learning, given lots of interesting applications in black-box attack, reinforcement learning, and fine-tuning language models.

**Weaknesses:**

1. The authors prove that ZSG converges with $tr(M)$, while previous work suggests that ZSC converges with $d\lambda(M)$. Based on this, the authors claim that ZSG is better when $tr(M)\leq d\lambda(M)$. I don't think this is a correct statement. There is no result in the paper showing that ZSC cannot achieve the rate $tr(M)$, and its $d\lambda(M)$ rate may come from the fact that previous analysis is not tight. To make the claim mathematically rigorous, the authors should provide the lower-bound under the current quadratic regularity assumption, showing that the rate of ZSC is $\Omega(d\lambda(M))$. Only then it is valid to say ZSG $\leq tr(M) \leq d\lambda(M) \leq$ ZSC.

2. I am also not sure why ZSC cannot achieve the rate $tr(M)$. The current ZSC considered in the paper queries all $d$ dimension at each iteration. Its complexity is thus deducted as $d$ times that of first-order methods. However, in ZSC, one can also only do random sampling at each iteration. For example, sampling from $\\{1,2,\cdots,d\\}$ instead of iterating over all $d$ dimension. This also builds a gradient estimator similar to ZSG, and similar analysis could apply. Specifically in the previous paper [Hanzely et al, 2018] and [Wang et al, 2024] mentioned by the authors, the rate of ZSC is also $tr(M)$ under the quadratic regularity assumption, e.g., Table 1 of [Hanzely, et al, 2018]. I am confused why authors say ZSC only achieves $d\lambda(M)$.

3. The rate of zeroth-order method has already been extensively studied under similar assumptions as the quadratic regularity assumption, e.g., [Malladi et al, 2023], [Yue et al, 2023], [arXiv: 2310.09639]. Stochastic mini-batch settings are also considered in these paper. Therefore, I am not sure how novel and challenge to obtain the results in the current paper given all these previous works.

4. The author claims in Corollary 4.4 that the algorithm will not converge with a fixed stepsize. I don't think this is correct. One can choose $\eta=(\log T)/T$, and then the algorithm converges with rate $T=(1/\epsilon)\log(1/\epsilon)$. Or can the authors clarify what they mean by a "fixed" step. In Corollary 4.6, when choosing $\sigma=0$, the complexity should reduce to the deterministic linear rate $\log(1/\epsilon)$. Is the current analysis tight?

5. I feel the paper is written in a rush and not well polished. There are lots of mistakes in grammar. For example, it should be smoothness assumption and strong convexity assumption in line 144; line 162-163 is not well written English; In Theorem 4.3, 4.5, it should be "let objective be quadratic" and "let x be update", etc.

**Questions:**

See weaknesses.

---

> ### Author Response · Authors · 2024-11-19
> **Rebuttal One**
>
> Dear Reviewer cWQZ,
> Thank you very much for your comments. We will address the weaknesses and questions in the following QnA format:
>
> Q1: The authors prove that ZSG converges with $ \mathrm{tr}(\mathbf{M})$, while previous work suggests that ZSC converges with $d\lambda(\mathbf{M})$. Based on this, the authors claim that ZSG is better when $\mathrm{tr}(\mathbf{M})\le d\lambda\mathbf{M}$. I don't think this is a correct statement. There is no result in the paper showing that ZSC cannot achieve the rate $\mathrm{tr}(\mathbf{M})$, and its $ d\lambda\mathbf{M}$ rate may come from the fact that previous analysis is not tight. To make the claim mathematically rigorous, the authors should provide the lower-bound under the current quadratic regularity assumption, showing that the rate of ZSC is $ \Omega(d\lambda(\mathbf{M}))$. Only then it is valid to say $ZSG\le \mathrm{tr}(\mathbf{M})\le d\lambda\mathbf{M} \le ZSC$.
>
> A1: There is a highly influential paper in the field of SGD that can provide some support. Considering the objective function $F$ is both strongly convex and smooth, Rakhlin et al. (2011) have already established the optimal convergence rate: $ \mathbb{E}[F(\mathbf{w}_T) - F(\mathbf{w}^*)] \leq \frac{2\mu G^2}{\lambda^2 T}.$ We calculate the partial derivatives in $d$ directions to obtain the gradient estimate, which can be directly extended to the optimal lower bound (Rakhlin et al., 2011). Our primary goal is to theoretically explain why ZSG outperforms ZSC in practice in most cases and why researchers tend to prefer the ZSG algorithm for model optimization. Many studies have adopted ZSG to fine-tune large language models, such as (Malladi et al., 2023), (Zhao et al., 2024), (Guo et al., 2024), (Chen et al., 2024), and so on. Our experiments also confirm the superiority of ZSG. This is because, for real-world datasets, the eigenvalue distribution of the Hessian is often skewed, meaning condition $\mathrm{tr}(\mathbf{M})$ ≪ $d \lambda\_{max}(\mathbf{M})$ holds. Our proposed theory can explain this phenomenon and provide valuable guidance for practical applications.
>
> Q2: I am also not sure why ZSC cannot achieve the rate $\mathrm{tr}(\mathbf{M})$. The current ZSC considered in the paper queries all dimension at each iteration. Its complexity is thus deducted as times that of first-order methods. However, in ZSC, one can also only do random sampling at each iteration. For example, sampling from $\\{1, 2, \ldots, d\\}$ instead of iterating over all dimension. This also builds a gradient estimator similar to ZSG, and similar analysis could apply. Specifically in the previous paper [Hanzely et al, 2018] and [Wang et al, 2024] mentioned by the authors, the rate of ZSC is also $\mathrm{tr}(\mathbf{M})$ under the quadratic regularity assumption, e.g., Table 1 of [Hanzely, et al, 2018]. I am confused why authors say ZSC only achieves $d\lambda(\mathbf{M})$.
>
> A2: These works are different from ours. First, Hanzely et al. (2018) and Wang et al. (2024) address different research problems compared to ours. Our study focuses on optimization problems in the finite-sum form, where at each iteration, we only need to access a subset of samples to construct the stochastic zeroth-order gradient estimate. In contrast, Hanzely et al. (2018) and Wang et al. (2024) require accessing the entire sample set at each iteration to construct their gradient estimates.
> Second, Hanzely et al. (2018) demonstrate that replacing Gaussian sampling with coordinate sampling, which corresponds to the commonly used coordinate descent method, can achieve the rate $\mathrm{tr}(\mathbf{M})$, provided that importance sampling technique is employed. However, in the context of zeroth-order optimization, it is not feasible to obtain information about the Hessian matrix, making it impossible to utilize importance sampling techniques.

---

> ### Author Response · Authors · 2024-11-19
> **Rebuttal Two**
>
> Q3: The rate of zeroth-order method has already been extensively studied under similar assumptions as the quadratic regularity assumption, e.g., [Malladi et al, 2023], [Yue et al, 2023], [arXiv: 2310.09639]. Stochastic mini-batch settings are also considered in these paper. Therefore, I am not sure how novel and challenge to obtain the results in the current paper given all these previous works.
>
> A3: We need to emphasize that these works are entirely different from ours. First, we outline a notable work that uses zeroth-order optimization algorithms to fine-tune large language models and highlight the differences from our contributions. Although Malladi et al. (2023) propose the descent theorem ($\mathbb{E}[\mathcal{L}(\boldsymbol{\theta}_{t+1}) | \boldsymbol{\theta}_t] - \mathcal{L}(\boldsymbol{\theta}_t) \leq -\eta \|\nabla \mathcal{L}(\boldsymbol{\theta}_t)\|^2 + \frac{1}{2} \eta^2 \ell \cdot \gamma \cdot \mathbb{E}[\|\nabla \mathcal{L}(\boldsymbol{\theta}; \mathcal{B})\|^2]$) for ZO-SGD, the ultimately proven global convergence rate $t = \mathcal{O} \left( \left( \frac{r}{n} + 1 \right) \cdot \left( \frac{\ell}{\mu} + \frac{\ell \alpha}{\mu^2 B} \right) \log \frac{\mathcal{L}(\boldsymbol{\theta}_0) - \mathcal{L}^*}{\epsilon} \right)$ is essentially that of gradient descent rather than stochastic gradient descent, as it includes a logarithmic term related to precision. Malladi et al. (2023) do not reveal the true convergence rate of ZSG! Yue et al. (2023) provide the convergence rate $ \mathcal{O} \left(\frac{ED_1}{\sigma_d}\log \left( \frac{1}{\epsilon} \right) \right) $ for standard zeroth-order optimization algorithm and the convergence rate $ \mathcal{O} \left( \frac{ED\_{\frac{1}{2}}}{\sqrt{\sigma_d}} \cdot \log \frac{L}{\mu} \cdot \log \left( \frac{1}{\epsilon} \right) \right)$ for accelerated zeroth-order optimization algorithm. Similarly, these are both based on gradient descent rather than stochastic gradient descent! The iterative algorithm $x\_{t+1} \gets x_t - \alpha \left( \frac{1}{n} \sum\_{i=1}^n \operatorname{clip}_C \left( \frac{f(x_t + \lambda u_t; \xi_i) - f(x_t - \lambda u_t; \xi_i)}{2\lambda} + z_t \right) u_t \right)$ proposed by Zhang et al. (2023) still relies on full gradient information rather than stochastic gradient information. Additionally, the vector $u_t$ used to construct the gradient estimate obeys Spherical distribution instead of Gaussian distribution. In summary, our theoretical analysis is entirely different from previous works. It does not require access to all sample information at each iteration, and the convergence rate $\mathcal{O} \left( \frac{\operatorname{tr}(M) \sigma^2}{\lambda\_{\min}^2(M)} \frac{1}{\epsilon} \right)$ we achieve is unique.
>
> Q4: The author claims in Corollary 4.4 that the algorithm will not converge with a fixed stepsize. I don't think this is correct. One can choose $ \eta = \frac{\log T}{T} $, and then the algorithm converges with rate $ T = \frac{1}{\epsilon} \log \left( \frac{1}{\epsilon} \right) $. Or can the authors clarify what they mean by a "fixed" step. In Corollary 4.6, when choosing $ \sigma = 0 $, the complexity should reduce to the deterministic linear rate $\log \left( \frac{1}{\epsilon} \right)$. Is the current analysis tight?
>
> A4: First, We apologize for the ambiguity in our statement. What we mean by a fixed step size is one that is independent of $T$. $\eta$ only needs to be a constant that satisfies relation $\eta \leq \frac{1}{12 \operatorname{tr}(M)}$. Then, for decreasing step size, it is unnecessary to assume $\sigma=0$ in Corollary 4.6 to recover the convergence rate of gradient descent. Because it is rare to gradually decrease the step size during gradient descent in practical scenarios. We primarily use mathematical induction to prove our conclusions, drawing inspiration from the optimal convergence analysis framework for stochastic gradient descent algorithms proposed by Stich (2019). Therefore, we believe that our analysis is rigorous.

---

> ### Author Response · Authors · 2024-11-19
> **Rebuttal Three**
>
> Q5: I feel the paper is written in a rush and not well polished. There are lots of mistakes in grammar. For example, it should be smoothness assumption and strong convexity assumption in line 144; line 162-163 is not well written English; In Theorem 4.3, 4.5, it should be "let objective be quadratic" and "let x be update", etc.
>
> A5: We sincerely appreciate your thorough review of our paper. We have made the requested revisions accordingly.
>
>
> References:
>
> [1]Alexander Rakhlin, Ohad Shamir, and Karthik Sridharan. Making gradient descent optimal for strongly convex stochastic optimization. arXiv preprint arXiv:1109.5647, 2011.
>
> [2]Sadhika Malladi, Tianyu Gao, Eshaan Nichani, Alex Damian, Jason D Lee, Danqi Chen, and Sanjeev Arora. Fine-tuning language models with just forward passes. Advances in Neural Information Processing Systems, 36:53038–53075, 2023.
>
> [3]Yanjun Zhao, Sizhe Dang, Haishan Ye, Guang Dai, Yi Qian, and Ivor W Tsang. Second-order fine-tuning without pain for llms: A hessian informed zeroth-order optimizer. arXiv preprint arXiv:2402.15173, 2024.
>
> [4]Guo W, Long J, Zeng Y, et al. Zeroth-Order Fine-Tuning of LLMs with Extreme Sparsity[J]. arXiv preprint arXiv:2406.02913, 2024.
>
> [5]Chen Y, Zhang Y, Cao L, et al. Enhancing Zeroth-order Fine-tuning for Language Models with Low-rank Structures[J]. arXiv preprint arXiv:2410.07698, 2024.
>
> [6]Filip Hanzely, Konstantin Mishchenko, and Peter Richt´arik. Sega: Variance reduction via gradient sketching. Advances in Neural Information Processing Systems, 31, 2018.
>
> [7]Yilong Wang, Haishan Ye, Guang Dai, and Ivor Tsang. Can gaussian sketching converge faster on a preconditioned landscape? In Forty-first International Conference on Machine Learning, 2024.
>
> [8]Pengyun Yue, Long Yang, Cong Fang, and Zhouchen Lin. Zeroth-order optimization with weak dimension dependency. In The Thirty Sixth Annual Conference on Learning Theory, pp. 4429–4472. PMLR, 2023.
>
> [9]Zhang L, Thekumparampil K K, Oh S, et al. DPZero: dimension-independent and differentially private zeroth-order optimization[C]//International Workshop on Federated Learning in the Age of Foundation Models in Conjunction with NeurIPS 2023. 2023.
>
> [10]Stich S U. Unified optimal analysis of the (stochastic) gradient method[J]. arXiv preprint arXiv:1907.04232, 2019.

---

> ### Comment · Reviewer_cWQZ · 2024-11-25
>
> Many thanks for the response!
>
> A1. I don't think the highly influential paper considers the quadratic regularity assumption. This means that the settings in the current paper and in the highly influential paper are different, and it is very likely that different optimal rates with different design and analysis techniques hold. My main point is that there is no theoretically sound explanation for why ZSC cannot achieve the Tr(M) rate.
>
> A2. According to (Hanzely et al., 2018), ZSC can achieve Tr(M) rate when $p_i\sim M_{ii}$ (Corollary 4.3.). I guess this is the Hessian information mentioned by the authors. However, in the analysis of ZSG in Theorem 4.5 in the current paper, the knowledge of Tr(M), $\lambda_{min}(M)$, and $\lambda_{max}(M)$ is also required to set up different parameters. This is also information about the Hessian matrix and is not feasible in the context of zeroth-order optimization.
>
> A3. Although these are different settings, similar analysis techniques can be used, which makes the contribution less significant.
>
> A4 and A5. Thanks for the explanation!

---

> ### Author Response · Authors · 2024-11-29
>
> Thank you for your reply!
>
> A.1 The strictly theoretical bound you described is not the main focus of our paper. Our core objective and primary contribution lie in attempting to explain, from a theoretical perspective, why ZSG demonstrates advantages in practice. This offers an insightful and innovative perspective, rather than merely conducting application-oriented research with ZSG in a conventional manner. We still believe that our work provides a sufficiently novel perspective.
>
> A.2 Hanzely et al. (2018) introduce importance sampling to achieve $ \mathrm{tr}(\mathbf{M})$ rate for ZSC. Their approach requires knowledge of the all diagonal elements of the Hessian matrix in order to accurately perform each iteration. The step size involved in our theorem is related to $ \mathrm{tr}(\mathbf{M})$, which aids in proving our result. In practice, although we do not know the exact value of $ \mathrm{tr}(\mathbf{M})$, we can improve practical performance by only adjusting step size (only related to $ \mathrm{tr}(\mathbf{M})$), which is not possible in (Hanzely et al., 2018). Additionally, $\lambda_{\min}(M)$ and $\lambda_{\max}(M)$ mentioned are only relevant to the proof process. In practice, there is no need to obtain these values.
>
> A.3 If you believe our contributions are not significant, then the paper ''Zeroth-order optimization with weak dimension dependency'', published in COLT, may also lack sufficient contribution. Additionally, in the field of fine-tuning large language models, Malladi et al. (2023) claim to prove the rate of ZO-SGD. However, what they actually prove is the rate of gradient descent (GD), without revealing the true rate of ZO-SGD. Based on your perspective, we believe this paper may not warrant publication either. We believe our work makes significant contributions and has the potential to provide valuable insights to the optimization community.

---

### Official Review · Reviewer_Gt8G · 2024-11-03

**Soundness:** 3
**Presentation:** 2
**Contribution:** 2
**Rating:** 3
**Confidence:** 3

**Summary:**

This paper investigates large-scale finite-sum optimization within the zeroth-order (ZO) stochastic optimization paradigm, focusing specifically on two methods: ZO-SGD-Gauss (ZSG), which pre-processes the stochastic gradient with a Gaussian vector, and ZO-SGD-Coordinate (ZSC), which estimates partial derivatives along coordinate directions. The study addresses the notable performance gap between ZSG and ZSC, aiming to provide theoretical insights that explain ZSG's empirically observed advantages. To achieve this, the authors introduce the "quadratic regularity assumption" on the Hessian matrix, a relaxation of typical smoothness and strong convexity assumptions. They demonstrate that this assumption allows for incorporating Hessian information into complexity analysis, yielding convergence rates that reveal ZSG's improved efficiency in certain settings. The authors validate their analysis through synthetic and real-world experiments.

**Strengths:**

The authors reinforce their theoretical findings with experimental results on both synthetic and real-world datasets, which enhances the paper's credibility. The empirical results are presented clearly and support the theoretical claims regarding convergence rates and query complexity.

**Weaknesses:**

The paper seems to be very interesting, however, the following points are present in the paper which hinder the perception of readiness and clarity of the paper:

- Introduction. The introduction is not well designed.... It is possible to improve this point, for example, a table where the result of the work will be clearly visible, as well as the efficiency compared to other algorithms.

- Could not find a link to github or other source where I can find the code of the experiments.

**Questions:**

See above

---

> ### Author Response · Authors · 2024-12-01
>
> Dear Reviewer Gt8G,
>
> Thank you very much for your feedback. We will address each of the concerns and issues you raised, and provide clarifications accordingly.
>
> 1.We have carefully considered your suggestions. Due to space constraints, we plan to incorporate a summary of our results and algorithm efficiency in the form of a table in the introduction section of the camera-ready version.
>
> 2.We are sharing a portion of the main code for our experiments, which can be accessed at the following link:https://anonymous.4open.science/r/222-3F2F.
>
> We would like to express our sincere gratitude once again for your valuable feedback！

---

> > ### Comment · Reviewer_Gt8G · 2024-12-03
> >
> > Dear Authors,
> >
> > Since my comments were not taken into account in the revised version of the paper, I am decreasing my grade to 3.

---

### Official Review · Reviewer_Styw · 2024-11-04

**Soundness:** 2
**Presentation:** 1
**Contribution:** 1
**Rating:** 3
**Confidence:** 4

**Summary:**

This paper provides a separation result between zeroth-order stochastic gradient descent and zeroth-order stochastic finite-difference method under a certain quadratic regularity assumption. The results essentially extends similar results in the deterministic setting to the stochastic finite-sum setting.

**Strengths:**

The paper address an important problem of providing separation results between two competing algorithms.

**Weaknesses:**

The main idea in the paper is (i) tr(M) ≪ d λ_max(M) and (ii) one algorithm has tr(M)  and the other has d λ_max(M), the complexity of the former algorithm is better than the latter. Without a formal lower bound for the latter, such a conclusion cannot be made.

Even ignoring this, similar results have been obtained in the deterministic setting previously and extension to the stochastic finite-sum setting is not significant and raise up to the level of ICLR acceptance.

**Questions:**

please see above

---

> ### Author Response · Authors · 2024-11-19
> **Rebuttal One**
>
> Dear Reviewer Styw,
> Thank you very much for your time and your comments on our work. We will address the weaknesses and questions in the following QnA format:
>
> Q1: The main idea in the paper is (i) $\mathrm{tr}(\mathbf{M})$ ≪ $d \lambda\_{max}(\mathbf{M})$ and (ii) one algorithm has $\mathrm{tr}(\mathbf{M})$ and the other has $d \lambda_{max}(\mathbf{M})$, the complexity of the former algorithm is better than the latter. Without a formal lower bound for the latter, such a conclusion cannot be made.
>
> A1: There is a highly influential paper in the field of SGD that can provide some support. Considering the objective function $F$ is both strongly convex and smooth, Rakhlin et al. (2011) have already established the optimal convergence rate: $ \mathbb{E}[F(\mathbf{w}_T) - F(\mathbf{w}^*)] \leq \frac{2\mu G^2}{\lambda^2 T}.$ We calculate the partial derivatives in $d$ directions to obtain the gradient estimate, which can be directly extended to the optimal lower bound. Our primary goal is to theoretically explain why ZSG outperforms ZSC in practice in most cases and why researchers tend to prefer the ZSG algorithm for model optimization. Many studies have adopted ZSG to fine-tune large language models, such as (Malladi et al., 2023), (Zhao et al., 2024), (Guo et al., 2024), (Chen et al., 2024), and so on. Our experiments also confirm the superiority of ZSG. This is because, for real-world datasets, the eigenvalue distribution of the Hessian is often skewed, meaning condition $\mathrm{tr}(\mathbf{M})$ ≪ $d \lambda\_{max}(\mathbf{M})$ holds. Our proposed theory can explain this phenomenon and provide valuable guidance for practical applications.
>
> Q2: Even ignoring this, similar results have been obtained in the deterministic setting previously and extension to the stochastic finite-sum setting is not significant and raise up to the level of ICLR acceptance.
>
> A2: We believe that the theoretical analysis in the stochastic finite-sum setting provides sufficient theoretical contributions to make our work acceptable to ICLR. We illustrate our point using a highly influential paper that applies zeroth-order optimization algorithms to fine-tune large language models as an example. Although Malladi et al. (2023) propose the descent theorem $\mathbb{E}[\mathcal{L}(\boldsymbol{\theta}_{t+1})|\boldsymbol{\theta}_t]-\mathcal{L}(\boldsymbol{\theta}_t) \leq -\eta \|\nabla \mathcal{L}(\boldsymbol{\theta}_t)\|^2 + \frac{1}{2} \eta^2 \ell \cdot \gamma \cdot \mathbb{E}[\|\nabla \mathcal{L}(\boldsymbol{\theta}; \mathcal{B})\|^2]$ for ZO-SGD, the ultimately proven global convergence rate $t = \mathcal{O} \left( \left( \frac{r}{n} + 1 \right) \cdot \left( \frac{\ell}{\mu} + \frac{\ell \alpha}{\mu^2 B} \right) \log \frac{\mathcal{L}(\boldsymbol{\theta}_0) - \mathcal{L}^*}{\epsilon} \right)$ is essentially that of gradient descent rather than stochastic gradient descent, as it includes a logarithmic term related to precision. Malladi et al. (2023) do not reveal the true convergence rate of ZSG! In the context of large-scale optimization problems, Malladi et al. (2023) fail to provide the true convergence rate of ZSG in the stochastic finite-sum setting. In other words, they do not successfully attempt to theoretically explain why ZSG performs better in practice. This highlights the challenging nature of our work while also demonstrating its potential to offer significant insights.
>
> References:
>
> [1]Alexander Rakhlin, Ohad Shamir, and Karthik Sridharan. Making gradient descent optimal for strongly convex stochastic optimization. arXiv preprint arXiv:1109.5647, 2011.
>
> [2]Sadhika Malladi, Tianyu Gao, Eshaan Nichani, Alex Damian, Jason D Lee, Danqi Chen, and Sanjeev Arora. Fine-tuning language models with just forward passes. Advances in Neural Information Processing Systems, 36:53038–53075, 2023.
>
> [3]Yanjun Zhao, Sizhe Dang, Haishan Ye, Guang Dai, Yi Qian, and Ivor W Tsang. Second-order fine-tuning without pain for llms: A hessian informed zeroth-order optimizer. arXiv preprint arXiv:2402.15173, 2024.
>
> [4]Guo W, Long J, Zeng Y, et al. Zeroth-Order Fine-Tuning of LLMs with Extreme Sparsity[J]. arXiv preprint arXiv:2406.02913, 2024.
>
> [5]Chen Y, Zhang Y, Cao L, et al. Enhancing Zeroth-order Fine-tuning for Language Models with Low-rank Structures[J]. arXiv preprint arXiv:2410.07698, 2024.

---

> ### Author Response · Authors · 2024-11-29
>
> Dear Reviewer Styw,
>
> We have taken into account all the comments you have made.
>
> If you agree that we managed to address all issues, please consider raising your grade to support our work. If you believe this is not the case, please let us know so that we have a chance to respond.
>
> With Respect,
>
> Authors

---

### Meta-Review · Area_Chair_mZfM · 2024-12-14

**Metareview:**

This paper studies stochastic zeroth-order optimization. Traditionally, zeroth-order stochastic algorithms are based on SGD, which using Gaussian smoothing to estimate the gradient using zeroth-order information. This is called ZSG method. This paper studies estimating partial derivatives along coordinate directions, which is called ZSC method. The authors claim that ZSC achieves a better complexity than ZSG. However, the reviewers found that this is not rigorously justified. To support this claim, the authors need to develop a lower bound for ZSG, but this is not developed in the paper.

**Additional Comments On Reviewer Discussion:**

Further discussed the novelty.

---

### Decision · Program_Chairs · 2025-01-22

Reject